# Natural Products Containing ‘Rare’ Organophosphorus Functional Groups

**DOI:** 10.3390/molecules24050866

**Published:** 2019-02-28

**Authors:** Janusz J. Petkowski, William Bains, Sara Seager

**Affiliations:** 1Department of Earth, Atmospheric, and Planetary Sciences, Massachusetts Institute of Technology, 77 Mass. Ave., Cambridge, MA 02139, USA; seager@mit.edu; 2Rufus Scientific, 37 The Moor, Melbourn, Royston, Herts SG8 6ED, UK; bains@mit.edu; 3Department of Physics, Massachusetts Institute of Technology, 77 Mass. Ave., Cambridge, MA 02139, USA; 4Department of Aeronautics and Astronautics, Massachusetts Institute of Technology, 77 Mass. Ave., Cambridge, MA 02139, USA

**Keywords:** P–N bond, phosphoramidate, *N*-phosphorylation, P–S bond, phosphorothioate, *S*-phosphorylation, P–C bond, phosphonate, phosphinate, phosphine

## Abstract

Phosphorous-containing molecules are essential constituents of all living cells. While the phosphate functional group is very common in small molecule natural products, nucleic acids, and as chemical modification in protein and peptides, phosphorous can form P–N (phosphoramidate), P–S (phosphorothioate), and P–C (e.g., phosphonate and phosphinate) linkages. While rare, these moieties play critical roles in many processes and in all forms of life. In this review we thoroughly categorize P–N, P–S, and P–C natural organophosphorus compounds. Information on biological source, biological activity, and biosynthesis is included, if known. This review also summarizes the role of phosphorylation on unusual amino acids in proteins (*N*- and *S*-phosphorylation) and reviews the natural phosphorothioate (P–S) and phosphoramidate (P–N) modifications of DNA and nucleotides with an emphasis on their role in the metabolism of the cell. We challenge the commonly held notion that nonphosphate organophosphorus functional groups are an oddity of biochemistry, with no central role in the metabolism of the cell. We postulate that the extent of utilization of some phosphorus groups by life, especially those containing P–N bonds, is likely severely underestimated and has been largely overlooked, mainly due to the technological limitations in their detection and analysis.

## 1. Introduction

All life on Earth relies on phosphorous-containing compounds in metabolism. Phosphate esters (C-O-P) in particular play a central, uniting role for all living organisms on Earth. They are not only crucial building blocks of genetic material (DNA and RNA) but also act as a main energy transfer medium for the great majority of the metabolic processes of the living cell. Phosphate esters also participate in a highly organized regulation of other metabolic processes (e.g., in protein phosphorylation and in signaling molecules such as cAMP and in enzyme cofactors). The importance and advantageous properties of phosphate esters are also discussed at length in the context of the origin of life on Earth and its early evolution [1,2,3]. Due to the dominating role of the phosphate esters in biochemistry and genetics it is not surprising that phosphate esters have been the central focal point of biological research for more than a century.

Perhaps it is therefore not surprising that for many years the critical roles of nonphosphate classes of phosphorous-containing biochemicals in maintaining cellular homeostasis were often overlooked. In addition to phosphate ester functional group, life also makes P–N (phosphoramidate) (Section 2), P–S (phosphorothioate) (Section 3), and P–C (e.g., phosphonate and phosphinate) (Section 4) linkages.

In this review we thoroughly categorize the P–N, P–S, and P–C understudied natural organophosphorus compounds (Throughout the paper, for consistency, all oxyacids are shown in their protonated form; we note however that under physiological conditions one or more of the hydroxyl groups attached to a phosphorus atom may be deprotonated). We provide a summary of their biological source, biological activity, and biosynthesis, if known. We also discuss the role of phosphorylation of unusual amino acids in proteins (*N*- and *S*-phosphorylation) as well as natural phosphorothioate (P–S) and phosphoramidate (P–N) modifications of DNA and nucleotides with an emphasis on their role in the metabolism of the cell. We challenge the commonly held notion that these nonphosphate organophosphorus functional groups are an oddity of biochemistry, with no central role in the metabolism of the cell. We postulate that the extent of utilization of some of the ‘noncanonical’ phosphorus-containing functional groups by life (e.g., P–N bonds) has been severely underestimated and largely overlooked. We hope that this review will stimulate further interest of the broad scientific community in these understudied classes of compounds.

## 2. Natural Products Containing a P–N Bond (Phosphoramidates)

Traditionally compounds containing a nitrogen–phosphorus bond are divided into two separate classes: I and II [4,5]. Compounds of class I are characterized by the presence of terminal P–OH group (all *N*-phosphorylated amino acids in proteins belong to this group as well as many small molecules e.g., phosphagen—*N*-phosphocreatine). On the other hand, compounds belonging to class II have accessible terminal P–NH_2_ group (for example, adenosine 5′-phosphoramidate) (Figure 1).

At the first glance, post-translational *N*-phosphorylation of amino acids in proteins (Section 2.2) and small molecules containing a phosphorus-nitrogen single bond (P–N bond) are relatively rare in biochemistry and number only around 55 among known natural products (see Section 2.1 for more information on small-molecule natural products containing a P–N bond). However, the apparent rarity of a P–N bond among natural products and PTMs might be caused by biased methods of isolation and/or identification. For example, it might be prudent to revisit the commonly repeated notion that the phosphoramidate P–N bond is only rarely utilized by life.

The phosphoramidate bond hydrolyzes quite readily in acidic conditions, which means that acid extraction of acidic chromatography solvents or column materials are liable to destroy such compounds. This might make the isolation or identification of phosphoramidates in the cell a challenge (see Section 2.2 for more information on *N*-phosphorylation of proteins and peptides). Recent advances in analytical techniques, especially protein mass spectrometry suggest that P–N bond biochemistry is critical to many core metabolic functions in all of life (discussed in Section 2, below). P–N bond-containing phosphagens are central to energy metabolism, and have been known for more than a century. Many *N*-phosphorylations of proteins have only recently been identified as a widespread and critical part of the physiology of the cell. Both chemical groups are shared among a multitude of evolutionary distant organisms. Below (in Section 2.1 and Section 2.2) we present an overview of the P–N biochemistry utilized by life and hypothesize that our current view on the extent of life’s utilization of the P–N bond chemistry might be greatly underestimated.

### 2.1. Small-Molecule Natural Products Containing a P–N Bond

Natural products containing P–N bonds belonging to classes I and II can be further categorized into two main, broad groups of molecules: phosphormonoamidates, containing one P–N bond and a much less numerous, and a second group of molecules called phosphortriamidates, containing three P–N bonds. Phosphordiamidates (with two P–N bonds) are not known to be produced by life. The only known instances of the phosphordiamidates occur as catalytic intermediates (e.g., in the Histidine Triad Nucleotide Binding (Hint) proteins (see Section 2.1.2 for more information on Hint proteins)) [6]. We discuss in detail all known small molecule natural products containing P–N bond below.

Phosphoramidate hydrazides: Three distinct natural phosphorus-containing hydrazides are known. All of them are phosphormonoamidates with antibiotic properties. Antibacterial natural product FR-900137 (**1**) was isolated from *Streptomyces unzenensis* from Japanese soil [7,8]. FR-900137 antibiotic is active against both Gram-positive and Gram-negative bacteria [8]. Two other phosphormonoamidate hydrazide antibiotics—fosfazinomycins A (**2**) and B (**3**)—were isolated from *Streptomyces lavendofoliae* No. 630 [9]. It is important to note that compounds **2** and **3** apart from being classified as phosphoramidates (P–N bond) also belong to a class of compounds called phosphonates (P–C bond) (for more information on phosphonates see Section 4.1).



The biosynthetic pathway of fosfazinomycins A and B was recently elucidated [10,11,12]. Careful isotope labeling experiments suggest that the N–N bond in P–N–N motif in fosfazinomycins originates from nitrous acid [12]. The nitrous acid (nitrite) is a key substrate that is utilized in conversion of l-aspartic acid to the intermediate hydrazinosuccinic acid. The exact mechanism of this conversion is not yet known [12]. A total of four different evolutionarily conserved enzymes are involved in sequential transfer of hydrazine moiety onto a glutamate side chain before its deposition into final natural product (Figure 2). The biosynthetic strategy utilized in the synthesis of the N–N bond in fosfazinomycins differs greatly from the biosynthetic mechanisms of other N–N bond-containing natural products known so far. The canonical approach to the biosynthesis of the N–N bond involves its direct formation on the scaffold of the final molecule [13,14]. The novel approach to the biosynthesis of the N–N bond shows an interesting example of the convergent evolution in utilization of difficult and reactive chemistry such as hydrazines.

Phosphosulfoximines: There are only few isolated reports on identification of the natural P–N phospho derivatives of sulfoximines (monoaza analogs of sulfones). Two such molecules—**4** and **5**—were isolated from a *Streptomyces* sp. [16,17]. All known natural sulfoximines have neurotoxic or antibiotic properties. For a detailed review on natural sulfoximines and other N–S bond-containing natural products see [18].



The only known natural phosphortriamidates are phosphosulfamates. Four natural phosphosulfamates are known: phaseolotoxins (**6**–**8**) and sulphostin (**9**) [18]. They are characterized by a distinct atom and bond composition. Both phaseolotoxins (**7, 8**), phaseolotoxin-active metabolite octicidin (**6**), and sulphostin (**9**) contain an N–S bond in an uncommon sulfamate group, bonded to an equally rare phosphortriamidate motif, leading to a ‘tract’ of five heteroatoms in a row. Phosphosulfamates were reviewed thoroughly recently [18,19] and will not be discussed here further.



Other phosphormonoamidates: Diphenyl cyclooctylphosphoramidate (**10**) was isolated from a dinoflagellate *Karenia brevis*. It appears that despite its unusual structure, for a natural product, compound **10** does seem to be a toxic metabolite of *Karenia brevis*, although it was noted that the possibility remains that it may be derived from an unsuspected contaminant in the artificial culture medium [20].



Phosphoramidon (**11**) and its stereoisomer talopeptin (**12**) (talopeptin differs from phosphoramidon only by a single stereocenter, from 6-deoxymannose in phosphoramidon to 6-deoxytalose in talopeptin) were isolated from cultures of various *Actinomycetes* species including *Streptomyces tanashiensis* and *Streptomyces mozunensis* MK2 [21]. Both compounds are inhibitors of the thermolysin enzyme and other metalloproteinases [22,23,24,25,26,27,28,29].



Another phosphoramidate antibiotic called JU-2 (**13**), containing two l-phenylalanine residues, two glucose residues, and linoleic and erucic fatty acid chains, was isolated from a widely-known *Streptomyces* species—*Streptomyces kanamyceticus* (*Streptomyces kanamyceticus* is an *Actinomycetes* species that the famous, widely-used, kanamycin antibiotic was originally isolated from) M8 [30].



The tunicate-associated bacterial strain of Streptomyces sp. JP90 found in the Great Barrier Reef (Australia), produces the structurally unprecedented metabolite cinnamoylphosphoramide (**14**) [31].



Several early studies identified a phosphophenylalanylarginine (FMPI) (**15**) from *Streptomyces rishiriensis* as a phosphoramidate antibiotic with in vitro inhibitory properties towards metaloproteases (with IC_50_ values measured in a nM range) [32,33,34,35,36].



The pathogenic bacterium *Campylobacter jejuni*, known for being a major cause of bacterial gastroenteritis worldwide, produces bacterial capsular polysaccharides (CPS) that are complex carbohydrate structures containing P–N bonds. CPS are composed of chains of sugars that surround the outer surface of the *Campylobacter jejuni* cells. CPS were shown to be instrumental in efficient colonization and infection of the host organism including defense from bacteriophages and host immune system [37,38,39,40,41]. A cluster of 35 genes is involved in biosynthesis and export of the CPS polysaccharides [42]. CPS are heavily chemically modified and many different *Campylobacter jejuni* strains produce strain-specific structural variations of the CPS [43,44]. The most unusual of the chemical modifications of the CPS is the incorporation of the unique *O*-methyl phosphoramidate modification on specific sugar residues (2-acetamido-2-deoxy-β-d-galactofuranose—**17**, d-glycero-α-l-gluco-heptopyranose—**18**) which so far have only been identified in *Campylobacter* sp. [43,44,45]. The biosynthetic pathway leading to the formation of the P–N bond in CPS was only recently discovered [46,47,48,49]. The initial transformation in the biosynthetic pathway for the phosphoramidate modification of the sugar residues in the CPS proceeds via direct *N*-phosphorylation of the amide nitrogen of l-glutamine with ATP by a specific l-glutamine kinase (EC 2.7.3.13) to form a *N*-phospho-l-glutamine (**16**) (Figure 3) [46,47,48,49].

#### 2.1.1. Phosphagens

Phosphagens are P–N containing compounds used as energy reserves in metabolism. We discuss them in a separate subsection due to their uniting chemical characteristics but also due to their convergent function in the molecular physiology of the cell.

Phosphagens’ main function in the cell is to alleviate an energy crisis when the demand for energy is higher than the cellular ATP production capacity. Phosphagens act as an ATP buffer, being made from ATP when ATP is abundant and being used to regenerate ATP from ADP when energy demand outstrips the cell’s capacity to synthesize ATP. Phosphagens can accumulate in the tissue to much higher intracellular concentrations than ATP [50]. In consequence the release of phosphate form phosphagen molecules in high-energy demanding tissues has an additional indirect regulatory effect on such basic, metabolic processes as glycolysis and glycogenolysis [50,51,52].

Phosphagens have been predominantly identified in the organisms belonging to the animal kingdom of life. The muscle tissue of animals requires high levels of freely accessible energy that has to be often released fast and in large amounts, this could lead to situations when ATP production is insufficient to meet the energetic needs of the animal muscle tissue (e.g., during fast rapid movements). However the phosphagen systems were also identified in single-celled organisms, like eukaryotic ciliates and flagellates [53,54,55] and even bacteria [53,54,55] (see below and Table 1 for an exhaustive list of phosphagens, their respective kinases, and organisms where they were characterized). The identification of phosphagen systems in unicellular organisms shows that phosphagens are used ubiquitously and possibly universally to mitigate physiological high energy demands. Such phosphagen-mediated fulfillment of high energy demands applies not only to complex tissues of multicellular organisms but also to individual organelles within the single cell. Phosphagen kinases of unicellular organisms are localized mainly in the ciliary region of the cell. Such localized concentration of phosphagens in the flagellar and ciliary regions likely enables an uninterrupted, continuous supply of energy to dynein and therefore enables a continuous ciliary movement [56,57]. It is generally proposed that arginine and creatine kinases were the ancestral phosphagen kinases (e.g. see work by Conejo [58]). However, our understanding of the evolutionary history of phosphagen systems is far from complete. For example, the discovery of phosphagen kinases (taurocyamine kinase and agmatine kinase), other than arginine kinase, in unicellular organisms (*Phytophthora infestans*, *Ochromonas danica*, and *Euglena oracilis*) signifies the complexity of the evolutionary history of these molecules [53,54,55].

The biosynthetic pathways differ depending on the phosphagen and the source organism, although the amino acid arginine is always the initial precursor for all phosphagens known to date [51]. The last step in the biosynthetic pathways of different phosphagens follows the same principle (Scheme 1).

In the phosphagen biosynthetic pathway proto-phosphagens are guanidine group-containing phosphate acceptor compounds. As a result of the phosphate transfer from the ATP molecule the P–N phosphoramidate bond is formed between the phosphorus atom and a nitrogen atom of the guanidine group, leading to the formation of the phosphagen product (Scheme 1). This final step of the phosphagen biosynthesis is catalyzed by specific phosphagen kinases. Phosphagen kinases are enzymes that catalyze the reversible Mg^2+^-dependent transfer of the gamma phosphoryl group of ATP to a naturally occurring guanidino compounds, proto-phosphagens, such as creatine, glycocyamine, taurocyamine, lombricine, and arginine. Phosphagen kinases are a highly conserved family of proteins that nevertheless differ significantly with respect to their enzyme specificity and protein structure (monomeric, dimeric, and oligomeric forms of phophagen kinases are known) and distribution in the cell [59,60,61,62,63,64,65,66,67,68,69,70]. Among the most studied phosphagen kinases is the creatine kinase (EC 2.7.3.2), which is the only known phosphagen kinase to exist in vertebrates [71,72,73,74]. In contrary to creatine kinase, the arginine kinase (EC 2.7.3.3) is most widely distributed, occurring in invertebrates [75] and even in several unicellular organisms [56,76], including strains of proteobacteria like *Desulfotalea psychrophila*, *Myxococcus xanthus*, *Moritella* sp. PE36, and *Sulfurovum* sp. NBC37-1 [53,54,55] (Table 1). It is still debated if bacterial arginine kinases identified in those few species are evidence of ancient evolutionary history of *N*-phosphoarginine and phosphoramidates or have been acquired by horizontal gene transfer from eukaryotes [53,54,55]. Other, less studied phosphagen kinases that were identified in invertebrates include hypotaurocyamine kinase (EC 2.7.3.6) [77], so far identified only in peanut worms [78], glycocyamine kinase (EC 2.7.3.1), lombricine and thalessemine kinases (EC 2.7.3.5), opheline kinase (EC 2.7.3.7), and taurocyamine kinase (EC 2.7.3.4) [51,79,80], identified mostly in annelids with several potential examples identified in trematodes [59,81,82,83,84,85,86,87,88,89] and unicellular oomycetes [51,79,80]. One phosphagen kinase, agmatine kinase (EC 2.7.3.10), appears to be specific only to protozoa [90] (Table 1).

It is notable that, as of yet, the phosphagen systems are not known to exist in plants, fungi, and in the majority of species of prokaryotic domains of life. Several studies attempted metabolic inverse engineering of *Escherichia coli* and *Saccharomyces cerevisiae* with the arginine kinase phosphagen system (both species are not known to produce phosphagens naturally) [91,92,93]. In both cases the engineered strains gained significant advantage in mitigating the adverse effects of low pH stress than their wild type counterparts [91,92,93]. It is conceivable that the extra boost of energy from the accumulated *N*-phosphoarginine (**19**) allowed for much more rapid recovery of arginine kinase-expressing strains of *Escherichia coli* and *Saccharomyces cerevisiae* [91]. These results show that at least in the case of *N*-phosphoarginine (**19**) the phosphagen system can be transferred directly to phylogenetically distant species by simple overexpression of arginine kinase with immediate beneficial effects for the modified organism [91].

There are nine phosphagens known. See Table 1 for an exhaustive list of phosphagens and the information on characterization of their respective kinases and organisms where they were originally identified.
*N*-phosphoagmatine (**24**) is a phosphagen identified in protozoa *Euglena oracilis* and *Ochromonas danica* [90]. As in the case of other phosphagens, a specific agmatine kinase (EC 2.7.3.10) is responsible for the synthesis of *N*-phosphoagmatine (**24**), with l-arginine also being phosphorylated but much less efficiently [90].*N*-phosphoopheline (**26**) is a phosphagen identified in the marine annelid *Ophelia neglecta*. The main function of the opheline kinase (EC 2.7.3.7) is the synthesis of the phosphagen *N*-phosphoopheline, however the substrate specificity of (EC 2.7.3.7) is much broader than other phosphagen kinases and (EC 2.7.3.7) can also phosphorylate taurocyamine, lombricine, and taurocyamine albeit with lower efficiency [94].*N*-phospholombricine (**25**) is a phosphagen identified in several invertebrate species, mostly annelids, e.g., earthworms [95,96,97,98]. The compound **25** is synthetized by the lombricine kinase (EC 2.7.3.5) which specificity varies with respect to the source species the enzyme was isolated from [63,99,100,101]. It is worth noting that different, evolutionarily distant, organisms can produce the same phosphagens that only differ in the stereoisomer of one component residue. For example, *N*-phospholombricine in majority of annelids contains a d-serine residue while the *N*-phospholombricine of echiuroids, a group of marine worms contains an l-serine moiety [51,79].*N*-phosphoguanidinoacetate (*N*-phosphoglycocyamine) (**23**) is a phosphagen identified in many invertebrate species, mainly annelids. The compound **23** is synthetized by the guanidinoacetate kinase (also named glycocyamine kinase; EC 2.7.3.1). Guanidinoacetate kinase participates in arginine and proline metabolism in the cell and is widely distributed across the invertebrate branch of the tree of life. The glycocyamine kinases (EC 2.7.3.1) from the annelid *Hediste diversicolor* was also shown to be responsible for the synthesis of the *N*-phosphoguanidine (**28**), however it is unclear if *N*-phosphoguanidine (**28**) is a true endogenous phosphagen of *Hediste diversicolor* or any other species [102]. While it is theoretically possible for *N*-phosphoguanidine to be formed in vivo and for compound **28** to be an important metabolite in the cell its importance in the cellular metabolism remains to be proven.*N*-phosphothalassemine (**27**) is a phosphagen structurally similar to lombricine. *N*-phosphothalassemine (**27**) was isolated from a common earthworm *Lumbricus terrestris* and an unsegmented marine worm *Thalassema thalassema* [103]. The phosphagen kinase EC 2.7.3.5, responsible for phosphorylation of lombricine, is also responsible for phosphorylation of methylated lombricines such as thalassemine [103].*N*-phosphohypotaurocyamine (**22**) is a rare sulfinic acid phosphagen so far identified only in peanut worms (*Golfingia sp.*) [78,104]. Hypotaurocyamine kinase (EC 2.7.3.6) responsible for the synthesis of *N*-phosphohypotaurocyamine has high preference towards hypotaurocyamine, although it can also phosphorylate taurocyamine, albeit with diminished efficiency [77]. It is suggested that this unusual phosphagen system evolved from molluscan *N*-phosphoarginine kinase [78].*N*-phosphotaurocyamine (**21**) is a sulfonic acid phosphagen synthesized by a widespread taurocyamine kinase (EC 2.7.3.4) identified in a large number of annelid species [59,81,82,83,84,85]. However, recent identification of taurocyamine kinases in a large number of non-annelid species, including trematodes *Paragonimus westermani*, *Schistosoma japonicum*, *Clonorchis sinensis* [82,83,86,87,88,89] and, in two isolated cases in unicellular oomycetes [82,83,86,87,88,89], suggests that the evolutionary and phylogenetic scope of alternative substrate specificities of phosphagen kinases may be more widespread than previously thought [105].*N*-phosphoarginine (**19**) phosphagen is as widespread among invertebrates as *N*-phosphocreatine (**20**) is widespread among vertebrate species. Arginine kinase (EC 2.7.3.3), responsible for phosphorylation of arginine, also occurs in unicellular organisms like protists and even bacteria which could suggest evolutionary ancient origins of *N*-phosphoarginine phosphagen system. Indeed, *N*-phosphoarginine phosphagen system appears to be the earliest one developed by life on Earth and at least couple of other phosphagen systems derive their evolutionary history from an earlier version of the *N*-phosphoarginine phosphagen system [51,54,58,75,105,106,107,108]. Interestingly d-arginine is a substrate for *Sabellastarte indica*d-arginine kinase [109,110].*N*-phosphocreatine (**20**) is a phosphagen synthesized by creatine kinase (EC 2.7.3.2). *N*-phosphocreatine (**20**) is produced both by vertebrates and invertebrates (Table 1). Compound **20** functions as a rapid reserve of high-energy phosphates to recycle ATP in high-energy demand tissues such as the brain or skeletal muscle and was mostly studied in mammals [111,112,113]. The chemical properties, functions, and clinical relevance of *N*-phosphocreatine (**20**) and its corresponding kinase were reviewed extensively elsewhere and will not be expanded here [106,114,115,116,117]. In brief, *N*-phosphocreatine is crucial for normal vertebrate physiology, not only on the whole organ level, e.g., in normal muscle activity, but also on the individual cellular level, e.g., in the formation of the ‘creatine kinase circuit’ that is essential for high-sensitivity hearing, as demonstrated by an unexpected hearing loss in creatine kinase knockout mice experiments [118].



After more than a hundred years of research focused on unraveling many functions of phosphagens in living organisms, it is clear that the P–N bond-containing guanidine compounds have rich and ancient evolutionary history and that they have a central role in the physiology and biochemistry of all high-energy demanding life, single- or multicellular.

Cyclic phosphoguanidines: We note that the structures of some phosphagens can undergo cyclization under acidic conditions to yield, e.g., glycocyamidine, creatinine, and their *N*-phosphorylated derivatives *N*-phosphoglycocyamidine (**29**) and *N*-phosphocreatinine (**30**) [112,243,244]. For example, it was postulated that 20–25% of the in vivo conversion of *N*-phosphocreatine (**20**) into creatinine may proceed via an intermediate, *N*-phosphocreatinine (**30**) [111,245]. However, the true physiological levels of *N*-phosphocreatinine are still uncertain. It appears that in rabbit white skeletal muscles the levels of *N*-phosphocreatinine (**30**) reach 0.4% of the levels of *N*-phosphocreatine (**20**) [111].

A compound structurally similar to *N*-phosphocreatinine (**30**) called dimethyl-*N*^2^-creatininylphosphate (**31**) was isolated from the sponge *Ulosa ruetzleri* collected from the inshore waters of Harrington Sound, Bermuda [246].



#### 2.1.2. Natural Phosphoramidate Nucleotides

Natural P–N bond-containing phosphoramidate analogs of common nucleotides are not widely studied. The most well-known natural P–N molecule belonging to the class II of phosphoramidates is adenosine 5′-phosphoramidate (AMPN) (**32**). Compound **32** is synthesized in vitro from adenosine 5′-phospho-sulfate (APS) and ammonia by an adenylyl transferase (EC 2.7.7.51) (Figure 4), an enzyme that is distributed in a wide variety of organisms including bacteria, algae, fungi and higher plants [247,248]. AMPN is in fact believed to be a core metabolite essentially present in all living organisms [249]. The adenosine 5′-phosphoramidate (**32**) was first isolated from the green alga *Auxenochlorella pyrenoidosa* [248] but was subsequently identified in cell extracts of many other organisms including *Escherichia coli*, *Dictyostelium discoideum*, *Euglena gracilis*, *Spinacia oleracea*, and *Hordeum vulgare* [247]. However, despite being identified in a multitude of species the function of AMPN is unknown.

AMPN can be hydrolyzed by at least two evolutionarily conserved enzymes dinucleoside triphosphatase (EC 3.6.1.29) and purine nucleoside phosphoramidase [250,251].



Apart from the identification of single phosphoramidate nucleosides, early studies also hinted at the existence of *N*-adenylylated proteins (**35**). It was suggested that *N*-adenylylated proteins might be very common [5]. Indeed early studies reported their discovery in several species (e.g., *N*-adenylyl proteins occur in membrane fraction of *Dictyostelium discoideum* [252,253]); however, their existence is not widely confirmed by more recent work. 

Early studies also postulated the existence of the phenylalanine adenylyltransferase (EC 2.7.7.54): an enzyme that catalyses the reaction of ATP with L-phenylalanine to produce pyrophosphate and *N*-adenylyl-L-phenylalanine in the biosynthetic pathway of the alkaloid cyclopeptin in *Penicillium cyclopium* [254,255]. However, since then, the existence of this enzymatic activity was questioned and currently BRENDA enzyme information system lists the enzymatic activity of EC 2.7.7.54 as part of EC 6.3.2.40, which does not involve formation of any products and intermediates containing P–N bonds [256]. More recent studies revised the biosynthetic pathway of cyclopeptin and further questioned the formation of a P–N bond metabolite during cyclopeptin biosynthesis [257].

Less controversial is the existence of *N*-adenylyl-l-lysine residues (**35**) either as a covalent enzyme substrate intermediates in DNA ligase (NAD^+^-dependent) (EC 6.5.1.2) in *Escherichia coli* and RNA ligase of wheat germ [258] or as substrates of the evolutionarily conserved Hint and Hint2 family of AMP/GMP phosphoramidate hydrolases (**36**, **37**) [259,260,261,262,263]. Hint proteins are enzymes catalyzing the purine phosphoramidate hydrolysis. They hydrolyze the P–N bond in substrate molecules containing an AMP/GMP-NH_2_ scaffold, for example, *N*-adenylyl-lysine residues (**35**), into AMP and a free amine (or ammonia) [263,264,265]. Hint proteins are highly conserved in evolution and are present in virtually every clade of life of Earth [266]. However, their exact biological role is not known. It is likely that the substrate specificity of AMP/GMP phosphoramidate hydrolases extends beyond lysine, towards other AMP/GMP P–N-linked amino acids [263,264].



The relatively more studied group of P–N bond-containing nucleotides is nucleotidic antibiotics. We review them below.

Canonical stearic and palmitic acid dinogunellins (**38**, **39**) and their structurally related putative cousins—dinogunellins A-D (**40**–**43**)—are adenosine-containing phospholipids. Originally, dinogunellins were detected in the mature eggs of several fish, e.g., in cabezon (*Scorpaenichthys marmoratus*) or northern blenny roe (*Stichaeus grigorjewi*), as well as in the blood of eels (*Anguilliformes*) [267,268,269]. The distinctive presence of a phosphoramidate P–N bond in dinogunellins is a recurring chemical feature that is widely shared among many nucleotidic antibiotics, e.g., phosmidosines, the antifungal nucleotide antibiotics from *Streptomyces durhameusis* (see below) [270,271]. In mature eggs of several fish dinogunellins occur as a lipoproteins, complexed with the egg protein vitellogenin [272]. Interestingly they are absent in immature eggs. The biological role of these compounds and the reason for their absence in the immature fish eggs is not known (apart from their toxic effect that could be used as a defense against predation).





Agrocin 84 (**44**) was originally isolated from *Agrobacterium radiobacter* K84 in Australia [273,274,275,276,277,278]. Agrocin 84 (**44**) is selectively active against several strains of the phytopathogenic agrobacteria like *Agrobacterium tumefaciens* and *Agrobacterium rhizogenes*. The toxic effect is achieved by inhibiting the tRNA synthetase in the pathogen [279,280,281]. Agrocin 84 is an *N*-6-phosphoramidate analog of an adenine nucleotide that contains 3-deoxyarabinose rather than ribose. The deoxynucleoside core, and the methylsubstituted pentanamide at the C-5 position of the deoxyribose moiety, are essential for its toxicity [275,278,282]. The 1-phospho-glucofuranose sugar moiety at the N-6 position is required for proper transport by susceptible bacteria but it is not in itself essential for agrocin 84 toxicity [282]. The genetics [283,284,285,286,287,288,289], biosynthesis, ecological and biogeographical context [287], specificity, and mechanism of toxicity of agrocin 84 was reviewed extensively before and it will not be covered here [290,291,292,293,294,295]. The toxic effect of the agrocin 84 system have been widely utilized in agriculture as a very effective biocontrol field agent against strains of *Agrobacterium tumefaciens* [289].



Apart from early reports on the isolation of agrocin 84 (**44**) and dinogunellins (**38**–**43**) mentioned above, only a handful of other phosphoramidate nucleotide antibiotics have been identified to date. A series of phosphoramidate nucleotide antibiotics called phosmidosines (phosmidosine (**45**), phosmidosine B (**46**), and phosmidosine C (**47**)) and two variants of *N*-methylphosmidosine (**48**, **49**) were isolated from the culture filtrate of *Streptomyces durhameusis* [271,296]. Phosmidosines appear to inhibit spore formation of *Botrytis cinerea* at the concentration of 0.25 µg/mL. *Botrytis cinerea* is a worldwide pathogenic fungus responsible for the grey mold disease in a variety of commercially important fruits and vegetables [271,296]. A series of isolated in vitro studies also suggested phosmidosines as potential anticancer agents [270,297,298,299].



Microcin C (Microcins C7 and C51 initially thought to be two distinct compounds were shown to be identical and are referred as microcin C) (**50**) and microcin C-like (e.g., **52**) peptide phosphoramidate antibiotics are members of a large family of ribosomally synthetized peptides that are produced by many species of bacteria, including cyanobacteria, both marine and terrestrial [300,301,302,303]. Virtually all of identified homologs of microcin C peptides contain a heptapeptide moiety together with the C-terminally attached adenosine phosphoramidate moiety (**50**). So far the only well studied exceptions from that rule are microcin C-like peptides from *Bacillus amyloliquefaciens* and *Yersinia pseudotuberculosis* which contain carboxymethyl-cytidine [304,305]. Natural antibiotics from the microcin C family are the so called “Trojan horse inhibitors” of aspartyl tRNA synthetases. The structure [306,307], genetics [308], biosynthesis [309,310,311,312], the antibacterial mechanism of action [313,314] and regulation of activity [315,316,317,318] of microcin C family of antibiotics was extensively reviewed before and will not be expanded upon here [319,320]. In short a peptide moiety (of variable length and amino acid composition depending on the organism of origin [303]) is responsible for the active transport of the inhibitory moiety part (phosphoramidate aspartyl-nucleoside, **51**, and **53**) into the bacterial cell. The phosphoramidate linkage is more stable to the hydrolysis, as compared to the labile native aspartyl-adenylate [314]. As a result the phosphoramidate “Trojan horse inhibitor” is capable of specific targeting of the aspartyl-tRNA synthetase and subsequent cessation of protein synthesis [314].



Until very recently all known homologs of microcin C peptide antibiotics (**50**) were characterized by relatively similar amino acid composition and length; all of them contained adenylate nucleosides. This relatively uniform picture of structural diversity of microcin C family of peptide antibiotics was questioned only very recently. It was shown that the microcin C-like peptides from *Bacillus amyloliquefaciens* and *Yersinia pseudotuberculosis* contain cytidine rather than adenosine [304,305]. Moreover, the microcin C-like peptide from *Yersinia pseudotuberculosis* (**52**) contains a much longer peptide moiety that is initially inactive and requires endoproteolytic processing inside producing cells. This postsynthesis processing is carried out by the evolutionary conserved TldD/E protease and is necessary for the antibiotic to achieve desired activity [305]. The result of the endoproteolytic processing is a peptide, 11-amino acid long, with C-terminal modified cytosine residue. The processed peptide is subsequently exported outside the producing cell (in its active form). The microcin C-like peptide from *Yersinia pseudotuberculosis* inhibits target cells in the same way as canonical microcin C from, e.g., *Escherichia coli*.



Apart from microcin C-like peptide antibiotics three other phosphoramidate pyrimidine nucleotide antibiotics are also known, including two hydroxylamines that contain a biologically unprecedented tract of six heteroatoms in a row (**54**, **55**). Antibiotics EM 2487 (**54**) and 1100-50 (**55**) were isolated from *Streptomyces sp*. Mer-2487 and *Streptomyces lavendulae* SANK 64297, respectively [321,322,323]. Antibiotics 1100-50 and EM2487 differ in their nucleoside substructure composition. EM2487 contains a uridine ring while 1100-50 has a cytidine instead. In addition, EM2487 contains an *N*-methylated hydroxylamine group, while the hydoxylamine group of 1100-50 is not *N*-methylated. Both EM2487 and 1100-50 were tested as potential anti-HIV and nematocide agents respectively [321,322,323]. The third known phosphoramidate pyrimidine nucleotide antibiotic—antibiotic HC 62 (**56**)—was isolated from *Bacillus sp*. HC-62 [324].



Although *N*-phosphorylation of exocyclic amino groups on the bases of nucleotides is seen in some of the compounds above (e.g., agrocin 84 (**44**)), direct *N*-phosphorylation of the nucleobases is not known in biology. The *N*-phosphorylated P–N bond-containing nucleotide ring variants of adenine, cytosine, and guanine where the amino groups of the base rings are phosphorylated are known to synthetic chemistry and were synthetized either by standard organic synthesis approaches [325] or by directed evolution of ribozymes [326]. We note that, so far, such modifications of single nucleotides were never observed in vivo nor were they identified on RNA or DNA. However, knowing that there are currently more than 160 different covalent modifications known to exist in RNA alone [327] the possibility of such unusual phosphorylations to occur in nature should not be excluded a priori.

### 2.2. N-phosphorylation of Proteins and Peptides

*N*-phosphorylation of proteins is a post-translational modification (PTM) that involves the addition of a phosphate group to a nitrogen atom of a basic amino acid side chains like l-arginine (Section 2.2.1), l-lysine (Section 2.2.2), or l-histidine (Section 2.2.3) and as a result the formation of a phosphoramidate P–N bond-containing PTM (for general overview of phosphoramidate modification of proteins see work of Besant, Attwood and others [328,329,330,331,332]).

The general acid lability of phosphoramidate PTMs in proteins meant that the progress on the identification of those protein modifications has been very slow and for decades only few scattered examples of P–N bond containing proteins from variety of organisms were known. Recent breakthroughs in biochemical and analytical methods of detection of *N*-phosphorylated proteins has resulted in the discovery of many more P–N bond containing PTMs, showing that the persisting view on the rarity of the P–N phosphoramidate bond in the core biochemistry of life is an artifact of the established, standard biochemical methods of detection (which are still often implemented today). For example, the standard purification and identification procedures for phosphor-proteins use an acid treatment. *O*-phosphorylated proteins are generally stable in acid conditions [333], but phosphoramidate proteins are not. Such bias in identification methods led to the *O*-phosphorylated proteins to be detected and, as a consequence, studied much more often than their *N*-phosphorylated counterparts. Hence over the years the *N*-phosphorylated proteins have been largely overlooked and the true extent of the utilization of the phosphoramidate P–N bond in cellular metabolism could not have been reliably assessed [328,331,334,335]. Likewise, the enzymes catalyzing the formation (kinases) and hydrolysis (phosphatases) of P–N bonds remained largely uncharacterized.

#### 2.2.1. *N*-phosphorylation of l-arginine

There are several sporadic, early reports describing protein arginine kinase activity, responsible for *N*-phosphorylation of specific arginine residues (**57**) in specific proteins. Those early studies particularly focused on identification of arginine *N*-phosphorylation in vertebrate tissues.



One of those early studies suggested *N*-phosphorylation of multiple arginine residues in histone H3 [336,337]. Interestingly, it was shown that the occurrence of the *N*-phosphorylated arginine residues on histone H3 (in rat heart endothelial cells) is dependent on the step of the cell cycle progression as the *N*-phosphorylation of arginine residues in histone H3 occurred only in quiescent cells and was absent in the dividing cell population (It is interesting to speculate if the observed cell cycle-dependent difference in *N*-phosphorylation of arginine residues in histone H3 has anything to do with *N*-phosphoarginine being used as another regulatory mechanism of chromatin condensation during cell cycle progression. *N*-phosphorylation of arginine converts the DNA-binding positive charge on arginine to a negative charge that abolishes such binding. Such a charge switch on arginine residues could be used as a signal in chromatin condensation) [337]. Another early study characterized an arginine-specific protein kinase tightly bound to rat liver DNA that was capable of autophosphorylation [338]. Several studies also suggested the essential role of protein arginine *N*-phosphorylation for the function of the viral protein VP12 involved in the viral replication cycle [339,340]. The VP12 protein is produced by granulosis virus infection of the Indian meal moth *Plodia interpunctella* [339].

In recent research, a growing body of experimental evidence suggests that arginine *N*-phosphorylation is an abundant protein posttranslational modification in both eukaryotes and prokaryotes. Recently, a combined mutational and mass spectrometric studies of protein tyrosine phosphatase B (PtpB)-deficient *Staphylococcus aureus* (PtpB is also an putative arginine phosphatase) showed that the number of putative arginine phosphorylation sites in proteome of *Staphylococcus aureus* alone exceeds 200 [341,342]. *N*-phosphoarginine post-translational modifications span proteins involved in virtually all metabolic processes of *Staphylococcus aureus*, from the energy metabolism and protein biosynthesis to the regulation of transcription and oxidative stress [341,342].

Only recently a detailed functional characterization of an arginine kinase (McsB protein arginine kinase (EC 2.7.14.1) from Gram-positive bacteria) was attempted [343,344]. For a short overview on the discovery of McsB protein arginine kinase see a summary by Suskiewicz and Clausen [345]. Protein arginine kinase McsB and its respective protein arginine phosphatase YwlE constitute a redox-sensitive kinase–phosphatase switch regulating the bacterial response to oxidative stress [346]. During oxidative stress, the two cysteine residues present in the active site of YwlE form a disulfide S–S bridge. The formation of the S–S bridge shuts down enzymatic activity of the phosphatase. The redox inactivation of YwlE leads to accumulation of *N*-phosphorylated arginine residues in proteins (due to uninterrupted activity of the McsB arginine kinase). For example, the accumulation of *N*-phosphorylated arginine residues leads to the inactivation of the transcription factor CtsR which under normal conditions represses expression of stress response genes in bacterial cells. The McsB-mediated *N*-phosphorylation of specific arginine residues in CtsR transcription factor prevents it from binding to the DNA, which in turn enhances the expression of the antistress response genes [343,347]. CtsR is likely not the only substrate for the McsB/YwlE kinase-phosphatase pair. Large numbers of *N*-phosphorylated arginine proteins were identified in *Bacillus subtilis*, many of them are transcription factors as well [348,349]. Similarly to results obtained for *Staphylococcus aureus*, *N*-phosphorylation of arginine residues in *Bacillus subtilis* proteins may be much more widespread and may play much more global role than previously assumed.

It is important to note that many of the putative or confirmed sites of arginine *N*-phosphorylation appear to participate in protein–DNA interactions. Arginine is one of the main amino acid residues that is involved in formation of protein–DNA interaction interfaces. One study suggests that as much as one-third of residues protein–DNA interfaces is arginine [350]. It logical to suggest that protein arginine *N*-phosphorylation could be an important means of regulating protein–DNA association. Being positively charged arginine is well suited for formation of tight electrostatic interactions with negatively charged phosphate backbone of the DNA. *N*-phosphorylation of arginine side chains allows for reverting of the net positive charge and therefore promoting the dissociation of proteins from their target DNA sequences. One may make an interesting general comparison between the three different charge states of arginine PTMs (*N*-methylation, conversion to citrulline, and *N*-phosphorylation) and speculate on the possible regulatory interplay between them [351]. While the *N*-methylation of arginine residues generally stabilizes the positive charge, the *N*-phosphorylation is not merely removing it (as it is in the case of a conversion to citrulline) but it reverts the charge, diametrically changing the properties of the modified residue.

Arginine is also crucial in mediating of many protein–protein interactions; thus, it is entirely possible that *N*-phosphorylation of arginine residues could also regulate protein–protein interactions, and not only in preventing such interactions, one can envision a mechanism in which a specific *N*-phosphorylated arginine residue works as molecular recognition platform for proteins that recognize such modified residues in a specific manner. Indeed at least one such case is known—ClpCP protease—which recognizes proteins with *N*-phosphorylated arginine residues and targets them for degradation. As shown by the recent study, arginine *N*-phosphorylation of proteins mediated by the McsB kinase is also involved in targeting of substrate proteins for degradation in Gram-positive bacteria [344]. Such complex regulation and diverse roles of this PTM further solidify the emerging picture of arginine *N*-phosphorylation as one of the central regulatory mechanisms in the physiology of the cell.

#### 2.2.2. *N*-phosphorylation of l-lysine

There are two different types of lysine *N*-phosphorylation that were detected as post-translational modifications (PTMs) on proteins. In both of them a phosphoramidate P–N bond is formed as a result of lysine *N*-phosphorylation. The first phosphoramidate PTM, where a single phosphate group is transferred to the nitrogen atom of the side chain of the target lysine residue, is called lysine *N*-monophosphorylation (**58**). The second described protein posttranslational modification involving lysine *N*-phosphorylation is called lysine *N*-polyphosphorylation (**59**), a covalent modification in which inorganic chains of polyphosphate are attached to lysine residues of target proteins. We describe both of them below.



Lysine *N*-monophosphorylation: Out of the three amino acids that were shown to be *N*-phosphorylated in vivo, *N*-monophospholysine (**58**) is the least studied. *N*-monophospholysine was initially described many decades ago in partially purified rat liver cell extracts [352,353]. While several early studies have suggested the presence of *N*-monophospholysine residues in proteins in vivo (e.g., lysine *N*-monophosphorylation in histone H1 [337,354,355]) and some studies even eluded to the identification of specific lysine kinases [356] and phosphatases [357,358,359,360], including PHPT1 histidine phosphatase (see Section 2.2.3 for more information on *N*-phosphorylation of histidine) that was shown to dephosphorylate chemically phosphorylated *N*-monophospholysines of histone H1 in vitro [361], the lack of reliable mass spectroscopy techniques to study this PTM have significantly limited our understanding of the function and the extent of this modification in the cell. The functions of *N*-monophosphorylation of lysine residues in a handful of known examples (e.g., the role of *N*-monophosphorylation of lysine in histone H1) is currently unknown [331]. However, recent advances in synthetic chemistry and mass spectrometric methods might allow for easier detection and characterization study of *N*-monophosphorylation of lysine residues in vitro and in vivo [362,363,364]. Some recent isolated studies suggest that the *N*-monophosphorylation of lysines could be a widespread phenomenon in biochemistry [365,366], indeed the apparent rarity of *N*-monophospholysine in biological systems might be only due to its notorious detection difficulty and not to the fact that *N*-monophosphorylated lysines are rare.

Lysine polyphosphorylation: The problem of robust detection of *N*-phosphorylated lysine residues is not limited solely to *N*-monophospholysines. It is accepted that only a very small fraction of lysine post-translational modifications have been identified experimentally [367]. Recent studies detected an unusual PTM associated with lysine residues present within acidic protein regions. Such lysine residues appear to have very long chains of inorganic polyphosphates covalently attached to the amino group of the lysine side chain (**59**) [332,368,369]. Once attached to the target lysine residue the inorganic polyphosphates polymer can reach a length of tens if not hundreds of phosphate monomer units [370,371]. A number of lysine *N*-polyphosphorylation targets have been identified in common budding yeast (*Saccharomyces cerevisiae*). The *N*-polyphosphorylated proteins in *Saccharomyces cerevisiae* include a nuclear signal recognition 1 (Nsr1) and its interacting protein topoisomerase 1 (Top1) [368], as well as 15 lysine *N*-polyphosphorylation protein substrates with functions related to ribosome biogenesis. *N*-polyphosphorylated proteins were also identified in human cells. For example, six PASK-domain containing target proteins were identified as targets for lysine *N*-polyphosphorylation [372]. The PASK-domain (poly acidic-, serine-, and lysine-rich sequence) appears to be a characteristic of *N*-polyphosphorylated proteins. The exact role of the PASK domains and *N*-polyphosphorylation is not understood although it is known that polyphosphates in general (not necessarily *N*-polyphosphates) participate in a variety of processes ranging from regulating core metabolism to structural roles [370]. It was recently suggested that polyphosphates are one of the key factors required for cell survival after DNA damage in eukaryotic cells [373]. It is therefore intriguing to speculate if *N*-polyphosphorylation also participates in these processes.

One of the most interesting possibilities is a potential interplay of *N*-polyphosphorylation of lysine residues with other PTMs that target lysine residues. For example, lysine sumoylation requires recognition sequence similar to the PASK domain. Lysine sumoylation could, in principle, target the same lysine residues and compete for them with *N*-polyphosphorylation [332]. Such complex interplay and competition between PTMs is not unheard of and was shown, e.g., for the *N*-terminal modifications of proteins (see below, Section 2.2.4 [374,375]).

Similar competition and interplay can also happen with serine pyrophosphorylation, even if *N*-polyphosphorylation and serine pyrophosphorylation are targeted towards different residues. Lysine *N*-polyphosphorylation is under indirect control of the levels of inositol pyrophosphates [368]. The most important function of the inositol pyrophosphate in the cell is the regulation of the levels of polyphosphates, and by extension the regulation of the levels of available ATP [376,377,378,379]. The connection between the inositol pyrophosphate and lysine *N*-phosphorylation opens the possibility of a complex interplay between lysine *N*-polyphosphorylation and serine pyrophosphorylation, because serine pyrophosphorylation is directly dependent on inositol pyrophosphate levels in the cell [380,381,382].

We conclude Section 2.2.2. by emphasizing that despite the fact that very little is known about the cellular functions of *N*-phosphorylation of lysine residues, recent breakthroughs in a variety of analytical techniques, especially mass spectrometry, are likely to revolutionize the *N*-phospholysine detection and characterization, hopefully also leading to an unambiguous detection of lysine kinases and phosphatases. So far, no in vivo specific lysine *N*-phosphorylating kinases nor *N*-dephosphorylating phosphatases have been identified [362].

#### 2.2.3. *N*-phosphorylation of l-histidine

The third of the phosphoramidate P–N bond-containing post-translational modifications (PTMs) of proteins known to exist is *N*-phosphorylation of histidine. The research on *N*-phosphorylation of histidine (and arginine and lysine as well) gained a lot of momentum recently with the rapid development of the new biochemical and analytical techniques aimed at thorough identification of the P–N bond modified amino acids in proteins. Development of specific anti-*N*-phosphohistidine antibodies, thanks to successful synthesis of acid-stable *N*-phosphohistidine analogs, and recent new mass spectroscopic approaches open the possibility for deeper understanding of the distribution of *N*-phosphohistidine. This understanding has in turn illuminated *N*-phosphohistidine biochemistry and cell biology, and its emerging roles in virtually all main cellular processes including cell cycle regulation, regulation of ion channel activity (e.g., in immune response), phagocytosis, or metal ion coordination (e.g., Cu(II)). As a result of the reinvigorated interest in *N*-phosphorylation of histidine, especially in mammalian systems, a series of excellent reviews were published that thoroughly cover the chemistry, identification and cellular functions of *N*-phosphohistidine, we will therefore only focus on a brief summary of the topic, referring the reader to the excellent recent literature summaries. For general reviews on the state of the *N*-phosphohistidine biology, see work by Klumpp, Besant, Fuhs and others [334,383,384,385,386] for its chemical properties and identification methods and see work by Attwood, Besant, Kee and others [329,335,387,388,389] for its enzyme-catalyzed formation (*N*-phosphohistidine kinases (EC 2.7.13.3), including the formation of *N*-phosphohistidine as a reaction intermediate in enzymes such as nucleoside diphosphate kinase (EC 2.7.4.6)) and hydrolysis (*N*-phosphohistidine phosphatases (EC 3.9.1.3)) of *N*-phosphohistidine residues see work by Besant, Attwood, Wieland and others [390,391,392,393,394].

The *N*-phosphohistidine as PTM exists in the form of two isomers 1- and 3-*N*-phosphohistidines (**60**, **61**), the third possibility 1,3-*N*’*N*’-diphosphohistidine (**62**) so far was not reported to be made by life, and may be unstable under physiological conditions [395].



Similarly to *N*-phosphoarginine and *N*-phospholysine, *N*-phosphohistidine is an acid-labile PTM [329]. It is only recently with the development of new approaches in detection of *N*-phosphohistidine and other *N*-phosphorylated amino acids that the true extent and the detailed roles of the phosphoramidate modification of proteins can be assessed.

It is now widely accepted that the histidine *N*-phosphorylation is a crucial component of cell signaling in all domains of life. The crucial role of *N*-phosphohistidine was first associated with regulation of rapid ‘canonical’ signaling pathways like the bacterial phosphoenolpyruvate–sugar phosphotransferase system (PTS), the reactions catalyzed by enzymes such as nucleoside diphosphate kinase and succinyl–CoA synthetase [396,397,398] or a well-studied ‘two-component’ system in bacteria and archaea [385,394,399]. Recently it became apparent that the *N*-phosphorylation of histidine residues is also a crucial component of many signaling pathways in eukaryotic cells, including fungi, plants, and animals, e.g., in the immune system in mammals [400,401,402]. Interestingly in mammalian proteins, the *N*-phosphorylation of histidine often happens at either 1-*N* or 3-*N* positions of the histidine imidazole ring (**60**, **61**), depending on the source of the kinase (EC 2.7.13.3), this of course adds to the complexity of the detection and identification of this modification in the mammalian cells [383]. The aberration in *N*-phosphohistidine homeostasis was also implicated in human diseases such as cancer or inflammation [403,404].

The overall occurrence of *N*-phosphohistidine in living organisms is currently estimated to be quite high, both in bacterial and archaeal cells as well as in eukaryotes. Recent studies suggest that *N*-phosphohistidine constitutes up to 10% of the total phosphorylation events in the eukaryotic cell [405], which makes the P–N bond-containing *N*-phosphohistidine 10 to 100 times more abundant than the well-studied *O*-phosphotyrosine (but less abundant than *O*-phosphoserine [334]).

Having established that the *N*-phosphohistidine is a central chemical modification of proteins in all of life, with hundreds of putative targets identified, the main focus of the *N*-phosphohistidine biology is directed towards identification of kinases and phosphatases that specifically regulate the *N*-phosphohistidine proteome. So far apart from a handful of prokaryotic kinases, only two mammalian histidine kinases (NME1 and NME2) [393] and three phosphatases (PHPT1 [406,407], LHPP [408], and PGAM5 [400]) were identified. It is certain that with the implementation of the new biochemical and analytical methods focused on identification of acid-labile phosphoramidate bond more *N*-phosphohistidine-dependent enzymes await discovery, not to mention the likely surge in the number of new confirmed *N*-phosphorylation targets.

#### 2.2.4. *N*-phosphorylation of Other Amino Acids?

The emerging view of *N*-phosphorylation of arginine, lysine and histidine as widespread post-translational modifications (PTMs) utilized by virtually all clades of life opens an interesting possibility for *N*-phosphorylation of other nitrogen-containing functional groups in proteins. Those potentially include *N*-phosphorylation of the nitrogen in the indole ring of tryptophan; *N*-phosphorylation of the amide nitrogen in asparagine and glutamine; or α-*N*-terminal phosphorylation of the α-amino groups on the *N*-terminal amino acid residues in proteins. We discuss the possibility of biological occurrence of those *N*-phosphorylated species below. 

The *N*-phosphotryptophan was successfully synthetized [409,410]. However, the *N*-phosphorylation of the indole ring of the tryptophan residue is so far unknown to occur naturally. In principle the *N*-phosphorylation of tryptophan residues is possible and was speculated on before [330]. The reasons for the apparent absence of the *N*-phosphorylated tryptophan can be two-fold. First, the bulkiness of the residue and its general hydrophobicity causes tryptophan side chains to be often buried deep within protein folds. This in turn might limit their overall accessibility towards putative *N*-phosphotryptophan kinases, not to mention the addition of a charged group to a buried hydrophobic residue could severely disrupt the overall protein fold. Secondly, it was also postulated that the required deprotonation of the indole nitrogen, before the phosphorylation reaction can occur, might be difficult to accommodate under physiological conditions [329,330].

The *N*-phosphorylation of l-asparagine or l-glutamine amide groups in proteins is also unknown. While the formation of the *N*-phospho-l-glutamine (**16**) was shown for the individual amino acid as a biosynthetic intermediate (see (Figure 3) and Section 2.1 for an in depth discussion of *N*-phospho-l-glutamine (**16**)) [47] the first *N*-phosphoasparagine or *N*-phosphoglutamine PTM still awaits discovery. The degree of physiological stability of such protein modification is also unknown.

Similarly, despite the fact that α-*N*-phosphorylated amino acids (both as free residues and in peptides) are widely-known to organic chemists, there have been no reports of α-*N*-phosphorylated amino acids in biological systems. This absence of α-*N*-phosphorylation of proteins is surprising. In the cell there is no shortage of free and accessible protein α-*N*-terminal amino groups that could potentially undergo such phosphorylation. 

Interestingly, many α-*N*-terminal PTMs are known to exist (for reviews see work by Tooley and Varland [411,412]), including α-*N*-terminal methylation [375,413,414], α-*N*-terminal acetylation [415,416], α-*N*-terminal propionylation [417], or α-*N*-terminal myristoilation/palmitoylation [418] to name a few, but the natural occurrence of α-*N*-terminal phosphorylation remains to be proven. There is also an intriguing possibility for such α-*N*-terminal modifications to be dynamic and interchangeable (on the same protein target) in a highly regulated manner [411]. Evidence for such interchangeability of α-*N*-terminal PTMs was recently suggested for myosin regulatory light chain 9 (MYL9), the first protein that can be either *N*-terminally acetylated or *N*-terminally methylated [374,375]. It is interesting to speculate if such complex regulation of α-*N*-terminal PTMs also, to any extent, involves α-*N*-phosphorylation. 

Such speculations do have merit, especially in light of the fact that α-*N*-terminally phosphorylated amino acids were postulated as one of the precursors in pre-biotic chemistry in several origin of life scenarios [419,420,421,422,423,424,425,426,427,428,429,430]. α-*N*-terminally phosphorylated amino acids can self-polymerize into oligopeptides under appropriate conditions. This chemical reactivity made them attractive candidates for possible intermediates in the prebiotic synthesis of the polymers of life [419,422,424,425,426,428,429]. Several pathways for the origin and coevolution of nucleic acids and proteins, involving α-*N*-terminally phosphorylated amino acids, were also proposed in an attempt to identify a common pre-biotic chemical building blocks for peptides and nucleic acids [419,421,425,426,428]. If such involvement of α-*N*-terminally phosphorylated amino acids at the origin of life indeed took place it is possible that some remnants of this chemistry still persist in modern biochemistry to this day.

However, so far despite all of the rich chemistry and possible intricate interplay between various α-*N*-terminal PTMs the biologically relevant occurrence of α-*N*-phosphorylation still remains to be discovered. It is very possible that, similarly to their *N*-phosphorylated side chain counterparts, α-*N*-phosphorylated peptides are acid-labile and transient regulatory species in the cell. If this is the case their detection would require specialized targeted-proteomics approaches, not unlike those applied to specifically identify *S*-phosphocysteine proteome (see work by Bertran-Vicente [431] and Section 3.2 for more information on *S*-phosphorylation of l-cysteine).

## 3. Natural Products Containing a P–S Bond (Phosphorothioates)

Interestingly natural products containing P–S double or P–S single bonds (phosphorothiones and thiophosphates respectively) are almost entirely absent form biochemistry. Apart from two instances of small molecule natural products (Section 3.1), *S*-phosphocysteine in several proteins (Section 3.2) and bacterial phosphorothioate DNA modification (Section 3.3) the P–S bond is not utilized by life. Such scarcity of P–S bond in biochemistry is in a direct contrast to a plethora of P–S compounds utilized in human industry. Phosphorothiones and thiophosphates are particularly popular as pesticides and herbicides. Compounds containing P–S bonds (double or single) are one of the most common functional groups found in chemicals utilized in agrochemistry. The number of phosphorothiones and thiophosphates utilized in agriculture alone is in the hundreds [432].

### 3.1. Small-Molecule Natural Products Containing a P–S Bond

Natural products containing a P–S bond are extremely rare in biochemistry. To date, there are only two reported cases, of chemically-related, P–S natural products known. The phosphorus-containing phosphorothione hydrazone (**63**) was isolated from the red tide dinoflagellate *Karenia brevis* near the coast of Florida and was identified as an ichthyotoxin [433]. It was also shown to exhibit acute toxicity towards rodents [434,435]. The second known small molecule natural phosphorothione is a dimerized form of the phosphorothione (**63**). Compound **64** was isolated from a marine fungus *Lignincola laevis* and also exhibits cytotoxic properties [436].



### 3.2. S-phosphorylation of Proteins and Peptides

In contrary to *O*-phosphorylated amino acids (like serine or threonine) which are generally quite hydrolytically stable at a broad range of pH values (The stability of different phosphorylated groups in acidic pH is as follows P–C > P–O > P–S > P–N. It is notable that phosphocysteine is more stable to acidic conditions than phosphoramidates such as phosphohistidine [330]) [437]; *S*-phosphorylated cysteine (**65**) is very prone to hydrolytic cleavage of the phosphothioester bond at acidic pH [330,431], e.g., at pH 3–4 the half-life of the phosphothioester bond does not exceed 15 min at 37 °C [330,437]. It is important to note as well that certain metal ions (like Cu(II) or Ag(I)) catalyze P–S bond hydrolysis [437]. At the first glance the hydrolytic lability of phosphothioester bond at acidic pH might be viewed as a serious obstacle to its use in biochemistry, and one of the reasons behind the apparent rarity of phosphothioesters in life. It is important to note however that *S*-phosphocysteine and other *S*-substituted phosphorothioic acids are very stable at basic pH (>7 up to 12). This chemical characteristic can open the possibility of utilization of stable *S*-phosphorylated cysteine residues in cell signaling or catalysis when the local cellular microenvironment (e.g., within the fold of the protein) is favorable. Indeed, as we briefly review below, the utilization of *S*-phosphocysteine in the regulation of a variety of cellular processes is likely much more wide spread than previously thought.



The single bond between phosphorous and sulfur (P–S bond) in *S*-phosphocysteine is widely-known to be present as a common enzymatic reaction intermediate and much less as a standalone post-translational modification of proteins. For example, widely reported instances of cysteine phosphorylation come from the studies of the catalytic mechanism of cysteine-dependent protein phosphatases (CDPs), especially protein tyrosine phosphatases (PTPs) [438,439,440,441]. PTPs catalyze the hydrolysis of phosphoester bonds with the formation of a *S*-phosphocysteine intermediate and are strictly specific towards phosphotyrosine-containing substrates. In PTPs a reactive, conserved catalytic cysteine thioate acts as a nucleophile and attacks the phosphorus atoms of the substrate phosphotyrosine [439,442]. The *S*-phosphocysteine intermediate catalytic state was detected in the crystal structure of a well-studied tyrosine phosphatase PTP1B (It is important to note that catalytic cysteine residue of PTP1B undergoes complex redox regulation, with the formation of interesting N–S bond-containing cyclic sulfenamide 1,2-thiazolidin-3-one [18,19]. Such redox regulation of the activity of catalytic cysteines protects them from detrimental effects of irreversible oxidation. More research is needed to uncover how does the cysteine *S*-phosphorylation fit within the broad redox metabolism of cysteine residues in the cell) [439].

It is only recently that *S*-phosphorylation of cysteine is recognized outside of its role as a catalytic intermediate and is identified more widely as a potentially important regulatory modification of proteins (for other recent overviews of this topic please see excellent work of Piggott, Attwood and Buchowiecka [330,437]). Recent reports that follow a series of early studies that identified *S*-phosphocysteine residue in the EIIB component of the phosphoenolpyruvate-dependent sugar transporter system in *Escherichia coli* [443,444,445,446] slowly add more examples of regulatory *S*-phosphorylation to the cellular repertoire.

For example, members of a transcriptional regulator family SarA/MgrA from *Staphylococcus aureus* undergo reverse phosphorylation of the conserved cysteine residue. The presence of the *S*-phosphocysteine in the SarA/MgrA transcriptional regulator was linked to the regulation of the virulence of the *Staphylococcus aureus* [447]. Apart from the *S*-phosphocysteine on the EIIB component of the bacterial phosphoenolpyruvate-dependent sugar transporter system SarA/MgrA (mentioned above) is so far the only confirmed instance of the *S*-phosphorylation of the regulatory cysteine residue.

The *S*-phosphorylation and *S*-dephosphorylation of SarA/MgrA regulatory cysteine residue is catalyzed by eukaryotic-like serine/threonine kinase (Stk1) and phosphoserine/phosphothreonine protein phosphatase (Stp1), respectively. Moreover, as shown by Sun and others [447], the *S*-phosphorylation of the cysteine residue in SarA/MgrA is inhibited by reactive oxygen species, due to cysteine oxidation. This observation is in agreement with the previous report that showed that SarA/MgrA transcriptional regulators are redox-regulated by reactive oxygen species released by the host immune system as a response to the microbial attack [448]. Thus, it appears that the connection between cysteine oxidation and *S*-phosphorylation requires a complex interplay between kinase/phosphatase activity regulation and the global cellular redox-state control. If *S*-phosphorylation of cysteine residues is routinely catalyzed by known serine/threonine kinases and phosphatases, as the SarA/MgrA case suggests, then the possibility that cysteine *S*-phosphorylation is much more common in the cell is quite likely.

In addition to the studies summarized above, one phosphoproteome analysis reported *S*-phosphorylation of Cys63 of rat heart sarcomeric mitochondrial creatine kinase [449]. The function the *S*-phosphorylation of this partially conserved residue is unknown.

In fact, novel mass spectrometric techniques aimed specifically at the detection of *S*-phopshocysteine were recently demonstrated [431]. The analysis reported by Bertran-Vicente [431] allows for a definitive identification of *S*-phosphocysteine in MS/MS spectra. Such technological breakthroughs open the possibility for the true assessment of the extent of utilization of *S*-phosphocysteine in cellular metabolism. It is likely that in the upcoming years regulatory *S*-phosphorylation of proteins will stop being treated as a rather obscure protein post-translational modification and will be recognized as one of the important regulatory signals in the metabolism of the cell.

### 3.3. Phosphorothioate DNA Modifications

The decades long, canonical view suggests that the basic repeating units of the DNA polymer are essentially composed of three residues: a purine or pyrimidine base, deoxyribose sugar, and a phosphate group, which are in turn composed of five (out of six) biogenic elements (hydrogen, carbon, oxygen, nitrogen, and phosphorus), leaving the sixth biogenic element—sulfur—out of the genetic material of life. Indeed, the absence of sulfur in the material injected by phage T2 into *Escherichia coli* cells was a Nobel Prize-winning part of the evidence that DNA and not protein was the genetic material [450]. This long-standing dogma was shattered with the relatively recent discovery of the phosphorothioation of bacterial DNA (e.g., **66**). The discovery of phosphorothioates (residues where the ionized, nonbonding oxygen atom of the phosphate group in the DNA backbone is replaced by sulfur) in bacterial DNA proved that sulfur’s role in building polymers of life is not solely limited to proteins. It appears that, at least in bacteria, sulfur is also crucial for maintaining normal functions of genetic material of life [451]. Contrary to all other known modifications of DNA and RNA, phosphorothioation of DNA is so far the only known natural modification of the phospho-sugar backbone of DNA.



The detailed functions, genetics, physiology, and biosynthesis of phosphorothioation of the bacterial DNA were reviewed very recently and will only be highlighted briefly here. For detailed, excellent review on this unusual modification please see work of Wang and others [452].

The prolific studies by Wang and colleagues identified phosphorothioation to be sequence-specific (requiring conserved core sequences d(G_PS_A), d(G_PS_G), d(C_PS_C), d(G_PS_T), d(A_PS_C), d(T_PS_C), and flanking sequences), stereospecific (R_P_ diastereoisomer configuration), as well as being postreplicative (i.e., the DNA is modified after it is synthesized). The phosphorothioate modification (PT) is regulated by the *dnd* operon gene cluster, consisting of five genes *dndABCDE* [453,454]. Phosphorothioation appears to be a very common phenomenon in phylogenetically diverse bacteria, with diverse sequence specificities and variable occurrence in different bacterial genomes [454,455]. So far, phosphorothioathion appears to be uniquely bacterial feature and was not detected in Eukaryotes and Archaea. See [456] and [452] for detailed summary of the genetics of the *dnd* cluster in various bacterial organisms.

Phosphorothioate DNA modification is synthetized by proteins DndA, C, D and E. It is postulated that the sulfur in PT modification is derived from desulfurization of cysteine by DndA. The rest of the synthetic mechanism is unknown.

As expected, and initially shown, the main PT function is as a component of the bacterial DNA restriction modification (r-m) defense system [457]. The bacterial r-m system also includes specialized endonucleases (type IV ScoMcrA) which specifically recognize foreign PT DNA [458]. Recent studies expanded the roles of PT DNA modification and showed that the cellular functions of phosphorothiation are much more versatile. It is now believed that PT is also involved in multiple cellular processes unrelated to defense against foreign DNA, including response to oxidative stress and balancing of the redox state of the cell [459,460] or involvement in epigenetic control of gene expression [461,462] and protection of DNA from double-stranded breaks [463]. For detailed information on the roles of PT in the bacterial cell see recent excellent review by Wang and others [452].

Endogenous phosphorothioate DNA restriction modification system was discovered only about a decade ago and many details regarding its genetics, enzymology and physiological function (both regarding bacterial defense mechanisms against foreign DNA invasion but also regarding many other newly discovered roles) are still unknown (see a review by Wang and others [452] for the most up to date state of research on PT DNA modification). We will conclude this section by outlining three outstanding questions regarding P–S bond formation in genetic polymers of life that in our view are the most fundamental and interesting ones:

1) Is PT modification of DNA a uniquely bacterial feature, or is it also present in Archaea and Eukarya? So far, no such modifications were detected in Archaean and Eukaryotic genomes.

2) DNA modifications are dynamic (e.g., DNA methylation and demethylation cycles crucial for normal functioning of the Eukaryotic cell). Is PT modification reversible or convertible in any way after implementation into bacterial DNA? So far there is no evidence for that and no enzyme responsible for removal of PT modifications was identified.

3) Is RNA is also modified by introduction of PT modifications? So far approximately 160 RNA modifications were discovered [327]. RNA is much more widely modified than DNA and so far, all the modifications identified in DNA were always eventually also found in RNA.

## 4. Natural Products Containing a P–C Bond

We conclude our overview of the natural products containing rare organophosphorus functional groups with a brief summary on the natural compounds containing a P–C bond. Phosphonates and phosphinates were studied extensively over the years and thoroughly reviewed very recently [464]. We will therefore cover phosphonate (Section 4.1) and phosphinate (Section 4.2) biochemistry only briefly, providing an overview of structural diversity of compounds belonging to these two classes of compounds. We conclude Section 4 with an overview of the rarely discussed and generally neglected topic of biochemistry of trivalent phosphorous compounds (Section 4.3). In Section 4.3 we review the extensive evidence that anaerobic life on Earth makes phosphine gas (PH_3_), a simplest trivalent phosphorous compound. We also explore the possibility that there may be other more complex phosphines made by life to be discovered.

For the ‘at-a-glance’ summary of the structural diversity of P–C bond containing natural products see list of molecules and their biological sources below: phosphonates (**67**–**117**), phosphinates (**118**–**131**), and phosphines (**132**, **134**).

### 4.1. Phosphonates

The P–C bond of phosphonates is one of the most stable bonds in biochemistry, much more stable than the analogous P–O bond. For example, 2-aminoethyl phosphonic acid (**72**), the first natural product containing a P–C bond to be discovered, is highly stable to hydrolysis, even by strong acids, highly thermally stable, and even stable to combustion [464]. Moreover, phosphonates are a class of chemicals known from their ability to inhibit many crucial biochemical pathways by mimicking phosphate monoesters and carboxylates in various metabolic processes. Such molecular mimicry of crucial metabolic intermediates is essentially possible due to two features: (1) highly chemically stable P–C bond and (2) a close overall structural resemblance of phosphonates to common biochemicals (e.g., phosphate esters or carboxylates). Those two features make phosphonates a great choice as targeted enzyme inhibitors, used both by nature (as natural antibiotics) and synthetic chemists alike.

At the first glance phosphonates can be considered an oddity of biochemistry. However, life on Earth appears to widely recognize phosphonates’ potential as carriers of useful biological function. A great variety of phosphonates have been isolated from virtually all clades of life: animals (including humans), fungi, plants, protozoans, archaea, and bacteria [464,465]. Natural phosphonates constitute both small molecule metabolites (see the structural overview of phosphonates (**67**–**117**) below) as well as phosphonylated macromolecules such as lipids, polysaccharides and glycoproteins (We note that, so far, despite many discoveries of diverse phosphonoglycoproteins a direct phosphonylation of proteins i.e. formation of a P–C bond-containing post-translationally modified amino acids in the peptide backbone is not known to exist) [465]. The wide extent of utilization of P–C bond-containing phosphonates by life is exemplified by the fact that in some organisms P–C compounds form the vast majority of cellular phosphorus-containing molecules (e.g., in eggs of a fresh water snail *Helisoma* 95% of phosphorus-containing compounds constitutes complex phosphonoglycans; in sea anemone *Urticina crassicornis* 50% is in the form of phosphonoglycans, phosphonoglycoproteins, and phosphonolipids; in *Tetrahymena pyriformis* 30% of membrane lipids is in the form of phosphonolipids [464,465]).

Therefore, phosphonates could be an underappreciated source of phosphorus in the global phosphorus cycle (for an excellent summary on the role of phosphonates in the global phosphorus cycle see work of McGrath [466]); especially in the marine environment where phosphorus is a main limiting nutrient. Recent studies on the cycling of phosphorus in the marine environment suggest that dissolved organophosphorus compounds are an important alternative source of the element for ocean life. A significant fraction (~25%) of the organophosphorus compounds dissolved in ocean water are compounds containing C–P bonds [467,468]. For example, the elemental composition of the cultured strains of cyanobacterium *Trichodesmium erythraeum* showed that up to 10% of the entire particulate phosphorous pool is contained in the form of P–C bond containing compounds (i.e., phosphonates) [469]. Since cyanobacteria are ubiquitous inhabitants of the marine environment it is not surprising that significant fraction of environmental phosphorus could be deposited in the form of phosphonates, which in turn can be utilized as a significant source of phosphorus by the rest of the marine biosphere. The importance of the P–C bond containing compounds, especially phosphonates in the global phosphorous cycle is supported by the metagenomic studies from the Global Ocean Survey on P–C bond biosynthesis and catabolism [470]. A significant number of marine microbial organisms appear to contain genetic clusters responsible for phosphonate biosynthesis (~10% of bacterial genomes studied) and coding for enzymatic pathways for hydrolysis of the P–C bond-containing compounds (~30% of collected bacterial genomes) [470].

As in the case of other rare organophosphorus natural compounds and biochemicals the true extent of the utilization of the P–C bond by life can only recently be properly assessed, in most part thanks to the multiple breakthroughs in environmental metagenomic techniques. Initial metagenomic studies on phosphonate biochemistry suggest that both their biosynthetic and catabolic pathways appear to be common and very diverse. Unfortunately, just like in the case of P–N bond-containing phosphoramidates, P–S bond-containing phosphorothioates or phosphines (see Section 4.3 below, for more information on the biochemistry of trivalent phosphorous compounds), the scarcity of suitable in situ detection, identification and quantification techniques for the P–C bond containing compounds hampers the progress in biochemistry of this important class of compounds.













### 4.2. Phosphinates

Phosphinates are a very rare class of natural products. The most well described group among them are several peptide antibiotics (**118**–**126**) containing a unique phosphinate amino acid phosphinothricin (**118**). They were isolated from various species of *Actinobacteria* including *Streptomyces viridochromogenes* and *Streptomyces hygroscopicus* (e.g., **123**) as well as *Kitasatospora phosalacinea* (e.g., **125**). The identification and biosynthesis of natural phosphinate antibiotics was reviewed before on many occasions and is not going to be expanded any further [464,465,471,472].





### 4.3. Phosphines

Organic compounds containing trivalent phosphorous are almost entirely excluded from biochemistry. The most studied example of biologically associated production of trivalent phosphorus-containing compound is that of a phosphine gas (PH_3_) (**132**). Phosphine is a simple volatile phosphorus compound. It is a reactive, irritating gas [473], a trace component of the Earth’s atmosphere [474]. Its production is associated with biological activity in a wide range of strictly anoxic (O_2_-free) environments [475,476]. For example, PH_3_ was detected in environments that include: lakes and rivers (whose bottom sediments are anaerobic) [477,478,479], biogas, and landfill gas that is a product of the anaerobic decomposition of domestic waste [480,481], a range of wetland and marshland soils [477,478,479]. All of such environments are complex, anaerobic locations with phosphate available as dissolved inorganic phosphate or as organophosphate-containing chemicals.

It was shown before that production of both phosphine and phosphite (another reduced form of phosphorus) by living organisms is thermodynamically plausible in reducing environments and at least some of phosphine production in the environment could be the result of energy capture reactions [482], suggesting that the direct production of phosphine by living organisms is likely if the environmental conditions are favorable.

Phosphine (as well as other reduced phosphorus species such as phosphite (**133**)) have been also found associated with feces and flatus from many animals including termites [483], penguins [484], cattle, pigs [485], and humans [486]. Guts are typically completely anaerobic, with anaerobic bacteria outnumbering aerobic bacteria [487,488,489], even in guts as small as those of earthworms [490] or termites [491].

Finally, several studies have reported phosphine production from mixed bacterial cultures in the laboratory [492,493]. One study claimed conversion of half the phosphorus in the culture medium (~180mg/L) into phosphine in 56 days [475]. The detailed biosynthetic pathway for microbial phosphine production is currently unknown. For detailed discussion of the biological production in the environment, its biochemistry, and atmospheric chemistry see three recent papers by Bains, Sousa-Silva and colleagues [482,494,495].



As stated above, compounds containing trivalent phosphorus are very rare in life, and so far, only one such compound other than phosphine has been identified. This compound is a cyclic alkyl phosphine phospholane (**134**), found in European badger (*Meles meles*) feces [496]. The detection of phospholane was carefully validated, and so it likely to be a genuine product of metabolism of anaerobic badger gut microbiome and not an anthropogenic contaminant. Feces are a highly anaerobic environment, so this detection supports the association of trivalent phosphorus metabolism with strictly anoxic environments.

We believe that there could be other biological trivalent phosphorus compounds awaiting discovery. The large majority of ‘natural products’ (i.e., chemicals made by life) are identified from aerobic samples—plants, animals, soil fungi, and *Ascomycetes* grown in aerated culture—as well as marine organisms [494]. Very few are collected from anaerobic samples, in part because of the received wisdom among natural product chemists that anaerobic organisms do not produce secondary metabolites [497]. The biological production of phosphine and phospholane shows that terrestrial life can indeed make trivalent phosphorus compounds, and suggests that it only does so in highly anoxic environments. We suggest that a more systematic search of the compounds made by organisms in completely anoxic environments will find other, more complex trivalent phosphorus compounds.

## 5. Conclusions and Future Directions

We have reviewed the occurrence, biological roles, and biosynthesis (if known) of nonphosphate biochemicals including P–N (phosphoramidate) (Section 2), P–S (phosphorothioate) (Section 3), and P–C (Section 4). We have also summarized the biochemistry of phosphorylation of ‘unusual’ amino acids in proteins (*N*- and *S*-phosphorylation), chemical modifications common in both prokaryotes and eukaryotes. We have also collected and summarized studies on the natural phosphorothioate (P–S) and phosphoramidate (P–N) modifications of DNA and nucleotides, a topic which is not widely covered in the literature.

Our review illustrates the diversity and centrality of the phosphorus metabolism in all life on Earth, and shows that life’s reliance on phosphorus goes well beyond phosphate. We further illustrate that fact by emphasizing the phylogenetic diversity of organisms making compounds containing nonphosphate phosphorus, showing that nonphosphate biochemicals are not a rare specialism of a few organisms but rather a crucial part of all of biochemistry.

Finally, our exhaustive review has shown that the extent of utilization by life of some of the ‘noncanonical’ phosphorus-containing functional groups (e.g., P–N bonds) was likely severely underestimated and largely overlooked over the years due to the technical difficulties associated with identification of nonphosphate natural compounds, meaning that their apparent rarity is probably an artifact. We hope that this review stimulates further research into this interesting class of natural compounds.

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
