# Peer review of "Natural Products Containing ‘Rare’ Organophosphorus Functional Groups"

_molecules, 2019, doi:10.3390/molecules24050866_

Round 1

Reviewer 1 Report

This was a true pleasure to read and superb review on a facet of living matter that is often forgotten about. I enjoyed the structural approach that the authors took and read it with enjoyment from head to tail. It is one of those pieces that rarely comes out but becomes a central piece to the field. This is not exaggerated.  I have provided a few suggestions for

Major Points

Add the stereochemistry: The authors need to add stereochemistry to each structure. The current convention in the natural product field is to NOT draw stereochemistry ONLY when it is unknown. According this review little if any stereochemistry is known. While many compounds have not been evaluated stereochemcally, and remained only with their 2D structures, several of the structures presented have know stereochemical centers. This information needs to be presented, as not presenting it is misleading. A good example is in Figure 2a where the name implies stereochemistry but as drawn it is not presented. This needs a major overhaul. While we understand why this may have been done, it needs to be revised so the presentation fits the currently accepted methods for drawing natural product structures. As shown in Figure 2, there are many different isomers that 18 could be as 5 stereocenters are unassigned.

Provide a proper reflection of the phosphorus bonds: This review is about unusual phosphorous bonds but most of the structure present very ugly and incorrect representations (even configurationally-impossible) of what the phosphorous bond would look like under physiological conditions.  For instance, N-phospho-L-glutamine is very poorly drawn and it only takes the reader a few seconds to search and find well drawn structures (https://www.ebi.ac.uk/chebi/searchId.do?chebiId=CHEBI:139506). In some figures, such as that used for 28-31, there three different representations of the phosphorous bond for four compounds. Unify the bond angles, and use a single representation throughout all figures. This even worse for conversions such as APS to 32 where the bond angles around the P change as a step happens. Moreover the entire structure flips from one geometric isomer to another. The authors really need to pay particular attention to each figure so the structural information remains consistent and shows the proper stereochemical arrangement. Many of the structures are not acceptable for publication.

The wavy lines need to be removed and the full structures need to be drawn. For instance it is very hard to know what is meant with the two wavely lines in 35? Clearly it can be inferred but as shown it is not correct and is rather misleading.  The authors should use chemdraw and check each structure minimally by its molecular formula before rendering a figure. Several of the structural errors would be immediately clear.

This reviewer can provide a detailed review of all the structures before it is published but as it stands I would have to comment on nearly every other structure with some issue from minor (bad bond angles) to major incorrect or missing structural units. The authors really need to redraw most of their 134 structures and add stereochemistry. Without this, this paper, while a joy to read, is not publishable.

Minor items:

One does not right ‘IC50’ rather ‘IC50 values’

On page 2 line 63, the authors start with compound 20 then 32. Typically, one goes numberically from 1 to ## in a manuscript. While this is a minor point they could put a figure in section 2 that shows the structure of 20 and 32 without the numbers and highlights the chemical bonding environment within them.  While I get that this does not matter, the reader has to dig a bit at this point. A small added figure with structures and names will do.

On page 3 line 97 and appearing generally within the manuscript. The convention is to use (2) when one gives a formal name and then drop the parentheses when just discussing the number alone.  This should be fixed generally so it fits with conventions one typically does not read ‘compound (2)’ but rather ‘compound 2’. This is an easy fix to do globally.

On page 3, line 99 and appearing generally within the manuscript. There is quite a bit of sending the reader off the read line in the manuscript such as (see section 4.1). While I understand the authors intent it may help to briefly state what the reader will see in section 4.1 such as (for more about P-C bonds see section 4.1).

The structures of 2 and 3 should be updated to reflect the recent discovery of Hammerschmidt (10.1039/C3OB41574K). This also goes for the biosynthetic pathway presented in Figure 1. The stereochemistry can be added to this figure, since the biosynthetic scheme does infer several of the chiral centers.

Figure 1. The authors should put dotted lines for the hypothetical steps not question marks. It is hard to see them and it is rather unconventional.

The structure of 5 has an OH and NH overlapping. The structure of 4 would be more conventionally drawn with the linear side chain like 5.  Again, the authors should comment on the stereochemistry of these materials.

The structure of 6-9 place the key Phosphosulfamate bonds in different configurations which detracts from the figure. This unit should be drawn the same for each ideally on the left or right and then the remaining structure follows suit. For a lack of a better term, the presentation looks sloppy.  Also one should fix the bond angles for instance while 8 is well drawn why is the key bond drawn so poorly? 

The structure of 10 is also crooked as the cyclooctyl ring is skewed.  While we understand this does not change the content it does detract from the presentation.

The structures of 11 and 12 are also missing that traditional right to left format. Clearly the configuration drawn would not exist as the two large sterical units are next to each other. A more conventional configuration can be drawn.  Like the other items above this is mostly cosmetic but as this manuscript is so well written a tad bit of editing of the structures could really make this an eye catching review.

There is an incorrect double bond on the upper right side of 13.  Also two hydroxyl groups overlap.

Why does the structure of 38-39 take an entire page. This is the case throughout this manuscript as the structures are rather disperse.

Bonds are missing such as the one within the adenylyl ring of 35. There are many this is only one example.

Table 1 can be compressed to fit on a single page or at the most two pages.

One does not capitalize compound names such as Microcin unless it is a commercial or product name. This should be fixed globally.

One only needs to write out Streptomyces once and there after use S. lavendulae with the exception of cases where the species was not described such that on line 477. 

Many of the abbreviations are not used enough and can be removed such as OOL on page 767.

Author Response

Response to Reviewer 1 Comments

This was a true pleasure to read and superb review on a facet of living matter that is often forgotten about. I enjoyed the structural approach that the authors took and read it with enjoyment from head to tail. It is one of those pieces that rarely comes out but becomes a central piece to the field. This is not exaggerated.  I have provided a few suggestions for

Response: We are very grateful for this reviewer’s supportive, detailed and very helpful comments. We hope that we have answered reviewer’s concerns adequately. Of course, we welcome any further suggestions if necessary.

If the reviewer agrees we would like to personally thank him/her in the acknowledgements section of the paper.

MAJOR POINTS:

Major Point 1: Add the stereochemistry: The authors need to add stereochemistry to each structure. The current convention in the natural product field is to NOT draw stereochemistry ONLY when it is unknown. According this review little if any stereochemistry is known. While many compounds have not been evaluated stereochemcally, and remained only with their 2D structures, several of the structures presented have know stereochemical centers. This information needs to be presented, as not presenting it is misleading. A good example is in Figure 2a where the name implies stereochemistry but as drawn it is not presented. This needs a major overhaul. While we understand why this may have been done, it needs to be revised so the presentation fits the currently accepted methods for drawing natural product structures. As shown in Figure 2, there are many different isomers that 18 could be as 5 stereocenters are unassigned.

Response 1: We agree with the reviewer and have updated the stereochemical information when known and appropriate.

Major Point 2: Provide a proper reflection of the phosphorus bonds: This review is about unusual phosphorous bonds but most of the structure present very ugly and incorrect representations (even configurationally-impossible) of what the phosphorous bond would look like under physiological conditions.  For instance, N-phospho-L-glutamine is very poorly drawn and it only takes the reader a few seconds to search and find well drawn structures (https://www.ebi.ac.uk/chebi/searchId.do?chebiId=CHEBI:139506). In some figures, such as that used for 28-31, there three different representations of the phosphorous bond for four compounds. Unify the bond angles, and use a single representation throughout all figures. This even worse for conversions such as APS to 32 where the bond angles around the P change as a step happens. Moreover the entire structure flips from one geometric isomer to another. The authors really need to pay particular attention to each figure so the structural information remains consistent and shows the proper stereochemical arrangement. Many of the structures are not acceptable for publication.

Response 2: We have standardized the representation of the structures throughout the manuscript. We have generally changed the ‘planar’ representation of the phosphorous-containing groups to ‘tetrahedral’ view.

We have tried to draw phosphates as suggested by the reviewer. In a few cases, however, such a layout on the page would result in crowding or overlap of other components of the molecule. In this case ChemDraw's 'clean up structure' function defaults to a ‘planar’ representation of the phosphate group, and we have retained this.

In some isolated cases (e.g. microcin or DNA phosphorothioate) we have left the ‘planar’ conformation of the phosphorous-containing groups for better clarity of the figure.

We have also generally unified the presentation, view and orientation of molecules presented in the manuscript (e.g. molecules presented in a single panel have similar conformation and orientations).

Major Point 3: The wavy lines need to be removed and the full structures need to be drawn. For instance it is very hard to know what is meant with the two wavely lines in 35? Clearly it can be inferred but as shown it is not correct and is rather misleading.  The authors should use chemdraw and check each structure minimally by its molecular formula before rendering a figure. Several of the structural errors would be immediately clear.

Response 3: The drawings of the molecules that contain ‘wavy lines’ represent specific cases of N-phosphorylated amino acids in a protein polymer and DNA phosphorothioate. The ‘wavy lines’ represent the fact that the residue shown is only one single monomer in a large polymeric chain and that the adjacent residues in the polypeptide or DNA chain, are not shown here.

We have added an explanation in each molecule figure that uses ‘wavy lines’ to make it more clear for the reader. Each structure was also drawn and checked with ChemDraw. 

Major Point 4: This reviewer can provide a detailed review of all the structures before it is published but as it stands I would have to comment on nearly every other structure with some issue from minor (bad bond angles) to major incorrect or missing structural units. The authors really need to redraw most of their 134 structures and add stereochemistry. Without this, this paper, while a joy to read, is not publishable.

Response 4: We are very grateful to the reviewer for the very detailed and thorough review and very valuable comments.

We have tried to correct all the structural errors, and apologise for letting them through to the submitted version. In all cases where a group of related molecules is shown in one panel or figure, we have made the depiction of phosphates (and sulphates) within that figure consistent, and also made the arrangement of other structural features within the figure as consistent as is practical to emphasise structural similarities

Within these constraints, we have tried to make sure all elements of structures are aligned with the page vertical and horizontal.

MINOR POINTS:

Minor Point 1: One does not right ‘IC50’ rather ‘IC50 values’

Response 1: Corrected.

Minor Point 2: On page 2 line 63, the authors start with compound 20 then 32. Typically, one goes numberically from 1 to ## in a manuscript. While this is a minor point they could put a figure in section 2 that shows the structure of 20 and 32 without the numbers and highlights the chemical bonding environment within them.  While I get that this does not matter, the reader has to dig a bit at this point. A small added figure with structures and names will do.

Response 2: We have added the requested figure, named it Figure 1 and re-numbered the remaining figures accordingly.

Minor Point 3: On page 3 line 97 and appearing generally within the manuscript. The convention is to use (2) when one gives a formal name and then drop the parentheses when just discussing the number alone.  This should be fixed generally so it fits with conventions one typically does not read ‘compound (2)’ but rather ‘compound 2’. This is an easy fix to do globally.

Response 3: We have corrected the numbering convention throughout the manuscript as requested by the reviewer.

Minor Point 4: On page 3, line 99 and appearing generally within the manuscript. There is quite a bit of sending the reader off the read line in the manuscript such as (see section 4.1). While I understand the authors intent it may help to briefly state what the reader will see in section 4.1 such as (for more about P-C bonds see section 4.1).

Response 4: We agree with the reviewer and we have made the references to different sections of the paper more informative. In instances where it was not immediately apparent we have added the information on the topic of the particular section that the reader is referred to. 

Minor Point 5: The structures of 2 and 3 should be updated to reflect the recent discovery of Hammerschmidt (10.1039/C3OB41574K). This also goes for the biosynthetic pathway presented in Figure 1. The stereochemistry can be added to this figure, since the biosynthetic scheme does infer several of the chiral centers.

Response 5: We have added the stereochemical information to structures 2 and 3 and to the biosynthetic scheme (and marked chiral centers on the chemical compounds). We have also cited Hammerschmidt’s work in the paper (as one of the information sources for the biosynthesis figure).

Minor Point 6: Figure 1. The authors should put dotted lines for the hypothetical steps not question marks. It is hard to see them and it is rather unconventional.

Response 6: Corrected.

Minor Point 7: The structure of 5 has an OH and NH overlapping. The structure of 4 would be more conventionally drawn with the linear side chain like 5.  Again, the authors should comment on the stereochemistry of these materials.

Response 7: We have added the stereochemical information when known and re-drawn the structures for clarity.

Minor Point 8: The structure of 6-9 place the key Phosphosulfamate bonds in different configurations which detracts from the figure. This unit should be drawn the same for each ideally on the left or right and then the remaining structure follows suit. For a lack of a better term, the presentation looks sloppy.  Also one should fix the bond angles for instance while 8 is well drawn why is the key bond drawn so poorly?

Response 8:  We have re-drawn the structures of 6-9 molecules and unified the drawing style of all the molecules presented in this figure. We have made the depiction of phosphoramidates (and sulfamates) within that figure consistent, and also made the arrangement of other structural features within the figure as consistent as is practical to emphasise structural similarities.

In this case we have made an exception however and presented the phosphoramidate and sulfamate groups in the ‘planar’ conformation for the clarity of the figure. It is possible to make the phosphoramidates and sulfamates ‘tetrahedral’, but only by making the lettering smaller / the bonds longer so groups do not overlap, and that presumably violates the Journal’s graphics standards.

Minor Point 9: The structure of 10 is also crooked as the cyclooctyl ring is skewed.  While we understand this does not change the content it does detract from the presentation.

Response 9: Corrected.

Minor Point 10: The structures of 11 and 12 are also missing that traditional right to left format. Clearly the configuration drawn would not exist as the two large sterical units are next to each other. A more conventional configuration can be drawn.  Like the other items above this is mostly cosmetic but as this manuscript is so well written a tad bit of editing of the structures could really make this an eye catching review.

Response 10: We have changed the conformation of compounds 11 and 12 to be more in line with the conformation suggested by the reviewer.

Minor Point 11: There is an incorrect double bond on the upper right side of 13.  Also two hydroxyl groups overlap.

Response 11: We have corrected the double bond in molecule 13 and changed the overall conformation of the molecule 13 to minimize functional group overlap.

Minor Point 12: Why does the structure of 38-39 take an entire page. This is the case throughout this manuscript as the structures are rather disperse.

Response 12: We have corrected the arrangement of molecules in the paper so they take less space. The final arrangement of molecules within a published pdf is going to be decided by the editorial office.

Minor Point 13: Bonds are missing such as the one within the adenylyl ring of 35. There are many this is only one example.

Response 13: We thank the reviewer for spotting this error. We have corrected the adenylyl ring of molecule 35 and another related molecule, 36, as well.

Minor Point 14: Table 1 can be compressed to fit on a single page or at the most two pages.

Response 14: As requested, we have compressed the information presented in Table 1 to fit approximately two pages.

Minor Point 15: One does not capitalize compound names such as Microcin unless it is a commercial or product name. This should be fixed globally.

Response 15: Corrected.

Minor Point 16: One only needs to write out Streptomyces once and there after use S. lavendulae with the exception of cases where the species was not described such that on line 477.

Response 16: If the reviewer agrees, we prefer to leave the full species names in the text. While we understand that the convention is to shorten the species name after the first use in the text, we came to the realization that it is easier to identify the exact species while reading the text if it is presented in its full name each time it is mentioned. It is especially beneficial in the reviews like ours which contain many species belonging to many different genera.  

Minor Point 17: Many of the abbreviations are not used enough and can be removed such as OOL on page 767.

Response 17: As requested, the following abbreviations were removed from the draft: OOL

Reviewer 2 Report

The manuscript of Petkowski at al. provides a very detailed and complete review about naturally occurring organophosphorus compounds with unusual linkage of phosphorous, particularly phosphorus being covalently attached to C, N or S. The manuscript presents all aspects of this topic and the literature is referenced very well. The manuscript is well written, clearly structured and easy to understand. Although there have been some reviews about that topic (many are referenced in the manuscript) this manuscript is clearly warranted and deserves publication.

I have only a few minor points that should be corrected:

1) In words like N-phosphorylation, O-methyl, N-phospho-L-glutamine, S-phosphocysteine etc. N, S and O should be written italic. Please correct that.

2) Names of amino acids: the authors write sometimes for instance L-phenylalanine and on other places phenylalanine (without L). They should be consistent and decide for one style; I would recommend writing L-...

3) The authors reiterate several times that the P-S, P-N and P-C compounds are underestimated. Although I agree with that statement, I think it is not necessary to repeat that several times.

4) Names of compound in figures are usually written small. However, compounds 34, 33, 34, 124, 126-128, 130-132 and 134 are written capital. A consistent style should be used.

5) Footnote 2: “L-Phenylalanine” should be corrected to “L-phenylalanine”.

6) Lines 414-473: agrocin 84 and microcin C are sometimes written capital, sometimes small; I recommend writing them always small.

7) Line 420: “N6” may be changed to “N-6” since that is more common.

6) Line 840: “E-coli” should be corrected to “E. coli”.

Author Response

Response to Reviewer 2 Comments

The manuscript of Petkowski at al. provides a very detailed and complete review about naturally occurring organophosphorus compounds with unusual linkage of phosphorous, particularly phosphorus being covalently attached to C, N or S. The manuscript presents all aspects of this topic and the literature is referenced very well. The manuscript is well written, clearly structured and easy to understand. Although there have been some reviews about that topic (many are referenced in the manuscript) this manuscript is clearly warranted and deserves publication.

Response: We are very grateful to this reviewer for this very positive feedback.

I have only a few minor points that should be corrected:

Point 1: In words like N-phosphorylation, O-methyl, N-phospho-L-glutamine, S-phosphocysteine etc. N, S and O should be written italic. Please correct that.

Response 1: Corrected.

Point 2: Names of amino acids: the authors write sometimes for instance L-phenylalanine and on other places phenylalanine (without L). They should be consistent and decide for one style; I would recommend writing L-...

Response 2: We thank the reviewer for spotting this. Where needed, we have changed the naming of amino acids in the paper to include L-, as requested.

We have also added “L-” to the titles of subsections about N-phosphorylation of amino acids in proteins and peptides to signify the fact that all information in a given subsection applies to L-amino acid.

Point 3: The authors reiterate several times that the P-S, P-N and P-C compounds are underestimated. Although I agree with that statement, I think it is not necessary to repeat that several times.

Response 3: We thank the reviewer for this comment. While we understand that it might not be necessary to repeat that statement as many times as we did in the paper, we have decided to leave the text unchanged. One of the main points of our paper is to show and emphasize the fact that the P-N, P-S and P-C compounds are not the rare oddity of biochemistry that they are often presented as. The repetition of the statements about the underestimation of those compounds throughout the text is designed to emphasize that fact and to remind the reader that all of those groups are generally underestimated. Hence the repetition of this statement in sections covering all classes of those compounds (i.e. P-N, P-S, P-C). 

Point 4: Names of compound in figures are usually written small. However, compounds 34, 33, 34, 124, 126-128, 130-132 and 134 are written capital. A consistent style should be used.

Response 4: Corrected.

Point 5: Footnote 2: “L-Phenylalanine” should be corrected to “L-phenylalanine”.

Response 5: Corrected.

Point 6: Lines 414-473: agrocin 84 and microcin C are sometimes written capital, sometimes small; I recommend writing them always small.

Response 6: Corrected.

Point 7: Line 420: “N6” may be changed to “N-6” since that is more common.

Response 7: Corrected.

Point 8: Line 840: “E-coli” should be corrected to “E. coli”.

Response 8: Corrected.

Round 2

Reviewer 1 Report

This is a massive improvement!!  The structures are much better. However, they still could use a bit of touch up and this paper will be ready for publication.

Figure 1, Rotate the Class II molecule about 5° clockwise so the bottom line of the pentose ring is on the horizon.  This should be done for all structures that are ‘skewed’

Figure 2. The structures flip around so that it is hard to map why the stereochemistry is inverting. The atoms should be in the same position going backwards so reorient L-Asp and hydrazinosuccinic acid so it matches the rest of the scheme and uses the same stereochemical projection.

Compound 4 has an odd bond length one atom away from the sulfur same with 5 but not as bad. This can be fixed easily

Compound 13 the author may note the sugar names predicted by the authors for JU-2

The structure of 14 is again skewed one can find a more appropriate representation when looking up cinnamic acid online

The structures of 17 and 18 are very confusing this is not add by their unusual angles. Please correct and align them so at least some bonds are up/down and not skewed.

Compound 21 the sulfonic acid is drawn with odd bond ages

Compounds 28-31 one OH is skewed. This is never done move the left OH up a bit and it fits fine.  Looks amateur.

The NH collides with the carbon skeleton in 38 and 39 same is true for the methyl group in 49.

It is more conventional to put R1 and R2 and show H or CH2CO2H for each isomer than what the authors did with spanning two bonds.

Compound 59 can be drawn in a left to right sense similar to 58.  The representation provided is rater complicated and hard to read. This can be fixed easily.

The authors should add the stereochemistry to 66.

The authors should check that structures 90-115 for missing stereochemistry as straight lines such as that in 110 or 116 indicate that these stereo centers remain unknown.

Author Response

Response to Reviewer 1 Comments

This is a massive improvement!!  The structures are much better. However, they still could use a bit of touch up and this paper will be ready for publication.

MINOR POINTS:

Minor Point 1: Figure 1, Rotate the Class II molecule about 5° clockwise so the bottom line of the pentose ring is on the horizon.  This should be done for all structures that are ‘skewed’

Response 1: Corrected.

Minor Point 2: Figure 2. The structures flip around so that it is hard to map why the stereochemistry is inverting. The atoms should be in the same position going backwards so reorient L-Asp and hydrazinosuccinic acid so it matches the rest of the scheme and uses the same stereochemical projection.

Response 2: As requested by the reviewer, we have unified the orientation of L-Asp, hydrazinosuccinic acid and N-acetyl-hydrazinosuccinic acid.

Minor Point 3: Compound 4 has an odd bond length one atom away from the sulfur same with 5 but not as bad. This can be fixed easily

Response 3: Corrected.

Minor Point 4: Compound 13 the author may note the sugar names predicted by the authors for JU-2

Response 4:  We have added the following text (bold) to the existing sentence about compound 13:

Another phosphoramidate antibiotic called JU-2 (13), containing two L-phenylalanine residues, two glucose residues, linoleic and erucic fatty acid chains, was isolated from a widely known Streptomyces species (…)

We have decided not to assign the stereochemical information to JU-2 due to lack of reliable information on the stereochemistry of this compound.

Minor Point 5: The structure of 14 is again skewed one can find a more appropriate representation when looking up cinnamic acid online

Response 5: We have corrected the orientation of compound 14 to resemble the orientation of cinnamic acid, as shown here: https://www.ebi.ac.uk/chebi/searchId.do?chebiId=CHEBI:27386

Minor Point 6: The structures of 17 and 18 are very confusing this is not add by their unusual angles. Please correct and align them so at least some bonds are up/down and not skewed.

Response 6: We have reoriented compounds 17 and 18 so that the bottom line of the sugar ring is on the horizon. We have also reoriented the phosphoramidate group for clarity and added an explanation for the use of ‘wavy lines’ as it is done for other polymeric structures throughout the manuscript.

Minor Point 7: Compound 21 the sulfonic acid is drawn with odd bond ages

Response 7: Corrected.

Minor Point 8: Compounds 28-31 one OH is skewed. This is never done move the left OH up a bit and it fits fine.  Looks amateur.

Response 8:  Corrected.

Minor Point 9: The NH collides with the carbon skeleton in 38 and 39 same is true for the methyl group in 49.

Response 9: We have moved the nucleoside residues of compounds 38, 39 and methoxy group of compound 49 a little bit to the left to minimize collision with the NH group.

Minor Point 10: It is more conventional to put R1 and R2 and show H or CH2CO2H for each isomer than what the authors did with spanning two bonds.

Response 10: We presume that the reviewer refers to compounds 40-43.

If the reviewer agrees we would like to retain our representation of compounds 40-43.

It is of course partially a personal preference to have the whole molecules shown, but also from the point of view of ‘big data’ chemoinformatics it is beneficial to have the whole structures shown in the paper. For e.g. modern computer software is capable of extracting molecule structures from figures from the pdf files if such figures are clearly depicted and are drawn in their entirety (the R1 and R2 convention is sometimes difficult to properly identify by automated algorithms). Therefore, presenting the structures in their entirety, and avoiding R1 and R2 convention, makes the task of software-assisted extraction of chemical structures much easier should such necessity arise.

Minor Point 11: Compound 59 can be drawn in a left to right sense similar to 58.  The representation provided is rater complicated and hard to read. This can be fixed easily.

Response 11: We have corrected the orientation of compound 59.

Minor Point 12: The authors should add the stereochemistry to 66.

Response 12: Corrected.

Minor Point 13: The authors should check that structures 90-115 for missing stereochemistry as straight lines such as that in 110 or 116 indicate that these stereo centers remain unknown.

Response 13: We have added the remaining stereochemistry to the P-C compounds if known.

Reviewer 2 Report

All questions were answered. I recommend publication of the manuscript in its current form.

Author Response

Once again we would like to thank the reviewer for favorable revision of our manuscript. 

We have answered all questions of this reviewer in round 1 of revisions.